# Lightning $NO_2$ simulation over the Contiguous US and its effects on satellite $NO_2$ retrievals

Qindan Zhu[1], Joshua L. Laughner[2,*], and Ronald C. Cohen[1,2]

[1]Department of Earth and Planetary Sciences, University of California, Berkeley, Berkeley, CA 94720
[2]Department of Chemistry, University of California, Berkeley, Berkeley, CA 94720
[*]Now in Department of Environmental Science and Engineering, California Institute of Technology, Pasadena, CA 91125

**Correspondence:** Ronald C. Cohen (rccohen@berkeley.edu)

**Abstract.** Lightning is an important $NO_x$ source representing ~10% of the global source of odd N and a much larger percentage in the upper troposphere. The poor understanding of spatial and temporal patterns of lightning contributes to a large uncertainty in understanding upper tropospheric chemistry. We implement a lightning parameterization using the product of convective available potential energy (CAPE) and convective precipitation rate (PR) coupled with Kain Fritsch convective scheme (KF/CAPE-PR) into Weather Research and Forecasting-Chemistry (WRF-Chem) model. Compared to the cloud top height (CTH) lightning parameterization combined with Grell 3D convective scheme (G3/CTH), we show that the switch of convective scheme improves the correlation of lightning flash density in the southeastern US from 0.30 to 0.67 when comparing against the Earth Networks Total Lightning Network; the switch of lightning parameterization contributes to the improvement on correlation from 0.48 to 0.62 elsewhere in the US. The simulated $NO_2$ profiles using the KF/CAPE-PR parameterization exhibit better agreement with aircraft observations in the middle and upper troposphere. Using a lightning $NO_x$ production rate of 500 mol NO flash$^{-1}$, the a priori $NO_2$ profile generated by the simulation with the KF/CAPE-PR parameterization reduces the air mass factor for $NO_2$ retrievals by 16% on average in the southeastern US on the late spring and early summer compared to simulations using the G3/CTH parameterization. This causes an average change in $NO_2$ vertical column density four times higher than the average uncertainty.

## 1 Introduction

Nitrogen oxides ($NO_x \equiv NO + NO_2$) are key species in atmospheric chemistry, affecting the oxidative capacity in the troposphere by regulating the ozone and hydroxyl radical concentrations (Crutzen, 1979). Anthropogenic sources (mainly fossil fuel combustion) are the largest contributor to the $NO_x$ budget on a global scale. Natural sources of $NO_x$ are also nonnegligible (Denman et al., 2007). While anthropogenic emissions of $NO_x$ are intensively studied, natural sources are less understood (e.g. Delmas et al., 1997; Lamsal et al., 2011; Miyazaki et al., 2012). Lightning contributes to ~10% of $NO_x$ budget on a global scale and represents over 80% of $NO_x$ in the upper troposphere (UT) (Schumann and Huntrieser, 2007; Nault et al., 2017). Over the US, the anthropogenic $NO_x$ emissions have been decreasing rapidly (Russell et al., 2012; Lu et al., 2015), making lightning an increasingly important source of $NO_x$ and an increasingly large fraction of the source of column $NO_2$. Ozone ($O_3$) in UT has long lifetime and leads to a more pronounced radiative effect than ozone elsewhere in the troposphere. Varying lightning $NO_x$

emission ($LNO_x$) by a factor of four (123 to 492 mol NO flash$^{-1}$) yields up to 60 % enhancement of UT $O_3$ and increases the mean net radiative flux by a factor of three (Liaskos et al., 2015). This range in the lightning $NO_x$ production rate is similar to the current uncertainty of estimated lightning emission rates. Further, incorrect representation of $LNO_x$ in a priori profiles for satellite $NO_2$ retrievals leads to biases in the retrieved $NO_2$ columns. This is exacerbated by the greater sensitivity of UV/Vis
$NO_2$ retrievals to the UT (e.g. Laughner and Cohen, 2017; Travis et al., 2016).

When lightning occurs, NO is emitted as a result of high temperatures and $NO_2$ forms through rapid photochemistry. Studies report the estimated $LNO_x$ production rate ranges widely from 16 to 700 mol NO flash$^{-1}$ (DeCaria et al., 2005; Hudman et al., 2007; Martin et al., 2007; Schumann and Huntrieser, 2007; Huntrieser et al., 2009; Beirle et al., 2010; Bucsela et al., 2010; Jourdain et al., 2010; Ott et al., 2010; Miyazaki et al., 2014; Liaskos et al., 2015; Pickering et al., 2016; Pollack et al., 2016;
Laughner and Cohen, 2017; Nault et al., 2017).

Two categories of methods, one emphasizing the near-field of lightning $NO_x$ and the other the far-field, have previously been applied to estimate $LNO_x$. In near-field approaches the total $NO_x$ from direct observation close to the lightning flashes is divided by the number of flashes from a lightning observation network to yield the $NO_x$ per flash (e.g. Schumann and Huntrieser, 2007; Huntrieser et al., 2009; Pollack et al., 2016). Near-field estimates of $LNO_x$ per flash have also been made
through use of cloud-resolved models with $LNO_x$ constrained by observed flashes and aircraft data from storm anvils (e.g. DeCaria et al., 2005; Ott et al., 2010; Cummings et al., 2013). In contrast, the far-field approach uses downwind observations to constrain a regional or global chemical transport model. The emission rate of lightning $NO_x$ is varied in the model (either ad hoc or through formal assimilation methods) until the modeled $NO_x$ agrees with the measurements of total $NO_x$ at the far field location (Hudman et al., 2007; Martin et al., 2007; Jourdain et al., 2010; Miyazaki et al., 2014; Liaskos et al., 2015; Laughner
and Cohen, 2017; Nault et al., 2017). In general, far-field approaches yield estimates of $LNO_x$ at the upper end of reported range, while estimates from the near-field studies are typically at the lower end of the range. Nault et al. (2017) showed that a large part of this discrepancy is because prior near-field studies assume a long $NO_x$ lifetime in the UT, while active peroxy radical chemistry in the near field leads to a short $NO_x$ lifetime (~3 h). Without accounting for this chemical loss, the near-field and far-field estimates are biased low compared to each other. However, this effect cannot completely reconcile the discrepancy
between $LNO_x$ reported from near- and far- field studies.

In chemical transport models, $LNO_x$ production is modeled by assuming a fixed number of moles of NO are produced per lightning flash, typically 250 or 500 mol NO flash$^{-1}$ (Zhao et al., 2009; Allen et al., 2010; Ott et al., 2010). This presents an additional challenge to the far-field approaches to constrain $LNO_x$, as errors in the simulation of lightning flashrate will propagate into errors in the $LNO_x$ production per flash. However, explicitly simulating the cloud scale processes that produce
lightning is generally too computationally expensive to be applied in a regional or global model as it requires spatial resolution at the scale of cloud processes. Instead, the convection is parameterized using simplified convection schemes. Lightning is then parameterized by a suite of convection parameters. The most prevalent lightning parameterization relates lightning to the cloud top height (CTH) (Price and Rind, 1992; Price et al., 1997). Price and Rind found a consistent proportionality between cloud-to-ground (CG) lightning flashes and the fifth power of cloud top height. Other meteorological variables, including
upward cloud mass flux (UMF), convective precipitation rate (CPR), convective available potential energy (CAPE), cloud ice

flux (ICEFLUX) have been suggested as alternative lightning proxies for CG flashes or in some cases total flashes (Allen and Pickering, 2002; Choi et al., 2005; Wong et al., 2013; Romps et al., 2014; Finney et al., 2014). When CG flashes are predicted, the total lightning rate, including CG and Intra-Cloud (IC) flashes, is derived by defining a regional dependent CG:IC ratio (Boccippio et al., 2002).

Several previous studies have evaluated the performance of these lightning parameterizations in regional and global models. Tost et al. (2007) concluded none of them accurately reproduce the observed lightning observations even though some are inter-comparable. Wong et al. (2013) showed that a model using the Grell-Devenyi ensemble convective parameterization and the CTH lightning parameterization simulates erroneous flash count frequency distribution over time while the integrated lightning flash count is consistent with the observation. Luo et al. (2017) tested the single-variable parameterizations (CTH,

CAPE, UMF, CPR) and the paired parameterizations based on power law relationship (CAPE-CTH, CAPE-UMF, UMF-CTH), each of which was coupled with Kain Frisch convective scheme, and demonstrated that the two-variable parameterization using CAPE-CTH improves upon the previous single-variable parameterizations; it captures temporal change of flash rates but the simulated spatial distribution is still not satisfactory.

In this study, we implemented the CAPE-PR lightning parameterization (Romps et al., 2014) into WRF-Chem and assess

the performance in reproducing lightning flash density. Our motivation is to produce a better representation of a proxy-based lightning parameterization in the regional chemistry transport model. We also evaluate the effect of modeled lightning $NO_x$ on both the a priori profiles used in satellite $NO_2$ retrievals and the retrievals themselves.

## 2   Methods: models and observations

### 2.1   WRF-Chem

This study applies the Weather Research and Forecast Model coupled with Chemistry (WRF-Chem) version 3.5.1 to the time periods May to June, 2012 and August to September, 2013. The model domain covers North America from 20 °N to 50 °N with 12 km×12 km horizontal resolution and 29 vertical layers. The North American Regional Reanalysis (NARR) provides initial and boundary conditions. Temperature, wind direction, wind speed and water vapor are nudged every 3 h towards to NARR product. Chemistry initial and boundary conditions are provided by the Model for Ozone and Related Chemistry

Tracers (MOZART, https://www.acom.ucar.edu/wrf-chem/mozart.shtml). Anthropogenic emissions are driven by the National Emissions Inventory 2011 (NEI 11), with a scaling factor to match the total emissions to 2012 emission from the Environmental Protection Agency (EPA, 2016). Biogenic emissions are driven by the Model of Emissions of Gases and Aerosol from Nature (MEGAN; (Guenther et al., 2006)). We use a customized version of the Regional Atmospheric Chemistry Mechanism version 2 (RACM2), the details are described by Zare et al. (2018).

The default lightning parameterization used in WRF-Chem is based on cloud top height (CTH). The parameterized lightning flash rates are proportional to a power of cloud top height with linear scaling varied by region:

$$f = \begin{cases} 3.44 \times 10^{-5} H^{4.9} & \text{Continental} \\ 6.20 \times 10^{-4} H^{1.73} & \text{Marine} \end{cases} \tag{1}$$

where $f$ is the CG flash rate in each grid and $H$ is the colocated cloud top height in units of kilometers.

We also implement an alternative lightning parameterization where lightning flash rates are defined to be proportional to the product of the convective available potential energy (CAPE) and precipitation rate (PR).

$$f = \begin{cases} 0.9 \times 10^{-4} \times E \times PR & \text{Southeastern CONUS} \\ 1.8 \times 10^{-4} \times E \times PR & \text{Elsewhere CONUS} \end{cases} \tag{2}$$

where $f$ the CG flash rate in each grid cell, $E$ the convective available potential energy and $PR$ the convective precipitation rate. Southeastern CONUS in the context is the region between 94 °W to 76 °W and 25 °N to 37 °N. This parameterization

was proposed by Romps et al. (2014). Romps et al. (2014) used a year-round observation of lightning and meteorological parameters and found a good correlation between observed lightning flash densities and observed CAPE times PR over the CONUS. CAPE-PR was further examined in Tippett and Koshak (2018) who computed the proxy in a numerical forecast model and found a fairly good agreement between the spatial pattern of the daily CG flash rate and the forecast proxy over 2003-2016. To our knowledge CAPE-PR parameterization has not previously been coupled with chemistry. Note that we

compute these two meteorological variables every 72 seconds in our model setup and produce lightning flash rates in a much shorter time step compared to Romps et al. (2014) and Tippett and Koshak (2018). We also apply a regional scaling factor of 0.5 to the southeastern US (See Sec 3.1).

    We analyze WRF-Chem outputs from three model runs. The first run, referred as "G3/CTH", is consistent with Laughner and Cohen (2017); it selects the Grell 3D ensemble cumulus convective scheme (Grell, 1993; Grell and Dévényi, 2002) and the

CTH lightning parameterization. The Grell 3D convective scheme readily computes the neutral buoyancy level which serves as the optimal proxy for cloud top height (Wong et al., 2013). The "G3/CTH" is the only option for the coupled convective-lighting parameterization used in WRF-Chem at a non-cloud resolving resolution (12 km). In addition, we run WRF-Chem with the CTH lightning parameterization coupled with the Kain-Fritsch cumulus convective scheme (Kain and Fritsch, 1990; Kain, 2004) ("KF/CTH") to test the effect of switching convective schemes. In the "KF/CTH" parameterization, the cloud top

height is the level where the updraft vertical velocity equals to zero. Another run, referred as "KF/CAPE-PR", selects the Kain-Fritsch cumulus convective scheme and the CAPE-PR lightning parameterization described above. Compared to the Grell 3D convective scheme, the Kain-Fritsch uses the depletion of at least 90% CAPE as the closure assumption and calculates CAPE on the basis of entraining parcels instead of undiluted parcels, which also improves the calculation of precipitation rate (Kain, 2004). The lightning $NO_x$ production rate is defined to be 500 mol NO flash$^{-1}$. The CG:IC ratio and the $LNO_x$ post-convection

vertical distribution are the same as used by Laughner and Cohen (2017).

## 2.2 ENTLN lightning observation network

To assess the performance of the lightning parameterizations we compare to lightning flashes from Earth Networks Total Lightning Network (ENTLN). ENTLN employs over 100 sensors across the United States and observes both CG and IC pulses (https://www.earthnetworks.com/why-us/networks/lightning/). All lightning pulses within 10 km and 700 ms of each other are grouped as a single flash. The IC and CG flashes are summed over the grid spacing defined in WRF-Chem.

Compared to National Lightning Detection Network (NLDN), ENTLN is selected for high detection efficiencies of both CG and IC flashes. The average detection efficiency for total flashes observed by ENTLN was 88% over CONUS relative to the space-based Tropical Rainfall Measurement Mission (TRMM) Lightning Imaging Sensor (LIS) (Lapierre et al. (submitted), private communication). Shown in Fig. S2, we matched the ENTLN data to LIS flashes both in time and space after the correction of LIS data based on its detection efficiency (Cecil et al., 2014) during May 13-June 23, 2012. It shows a median correlation ($R^2 = 0.51$) with the slope of 1.0, indicating the ENTLN data during the study time period is in agreement with the LIS observation. We use the ENTLN for analysis as reported and consider the detection efficiency of ENTLN as a source of uncertainty when comparing the modeled lightning flashes.

## 2.3 In Situ Aircraft Measurements

We compare our simulations to observations from aircraft campaigns that focus on deep convection. The Deep Convective Clouds and Chemistry (DC3) campaign (Barth et al., 2015) took place during May and June of 2012 over Colorado, Oklahoma, Texas and Alabama. The Studies of Emissions and Atmospheric Composition, Clouds, and Climate Coupling by Regional Surveys (SEAC4RS) (Toon et al., 2016) took place during August and September of 2013; most of the flight tracks occurred over the southeastern US. Both aircraft campaigns flew into and out of storms and sampled deep convection. The combination of these two aircraft campaigns cover the regions with the most active lightning in the domain.

## 2.4 Satellite Measurements

The Ozone Monitoring Instrument (OMI) is an ultraviolet/visible (UV/Vis) nadir solar backscatter spectrometer launched in July 2004 on board the Aura satellite. It detects backscattered radiance in the range of 270-500 nm and the spectra are used to derive column $NO_2$ at a spatial resolution of 13 km×24 km at nadir (Levelt et al., 2006). The OMI overpass time is ~13:30 local time.

We use the Berkeley High Resolution (BEHR) v3.0B OMI $NO_2$ retrieval (Laughner et al., 2018). The air mass factor (AMF) is calculated based on the high spatial resolution a priori input data including surface reflectance, surface elevation and $NO_2$ vertical profiles. In this study we apply an experimental branch of the BEHR product which differs from v3.0B in several ways. First, instead of calculation based on temperature profiles from WRF-Chem, the tropopause pressure is switched to GEOS-5 monthly tropopause pressure which is consistent with NASA Standard Product (SP2) (Mak et al., 2018). Analysis shows the algorithm used in BEHR v3.0B to calculate the WRF-derived tropopause pressure is very much dependent on the vertical

|  |  | G3/CTH | KF/CTH | KF/CAPE-PR |
|---|---|---|---|---|
| Southeastern | Slope | 2.08 | 0.94 | 0.96 |
|  | $R^2$ | 0.30 | 0.67 | 0.72 |
| Elsewhere | Slope | 0.98 | 0.54 | 1.19 |
|  | $R^2$ | 0.27 | 0.48 | 0.62 |

**Table 1.** Correlation statistics between observed and modeled (G3/CTH, KF/CTH, KF/CAPE-PR) flash density per day averaged by regions

spacing predefined in WRF-Chem setup, which causes biases when the vertical layers are at a coarse resolution. Second, the $NO_2$ vertical profiles are outputs using the modified lightning parameterization described in Eq. 2.

## 3    Results

### 3.1    Comparison with observed lightning flash density

The lightning parameterizations are compared against observations from ENTLN in Fig 1. Each of the datasets is averaged from May 13 to June 23, 2012, covering DC3 field campaign. The ENTLN data is summed to the $12 \text{ km} \times 12 \text{ km}$ WRF grid. The G3/CTH parameterization fails to reproduce the spatial pattern of flashes observed by ENTLN over the CONUS. Compared to the G3/CTH, the KF/CTH parameterization improves the spatial correlation in the southeast region of US and yields a lower amount of lightning flashes. It indicates that KF convective scheme produces smaller cumulus cloud top heights

than G3 scheme by including entrainment and detrainment processes during the convection. The result is consistent with Zhao et al. (2009). The KF/CAPE-PR parameterization better captures the spatial distribution of flash densities both in the southeast region and elsewhere in CONUS. However the KF/CAPE-PR parameterization still fails to capture the gradients in flash occurrence within smaller regions. For instance, ENTLN shows that more lightning occurs along the east coast than west coast in Florida, however, WRF-Chem generates a lightning flash density of the same magnitude over both areas. Nevertheless,

the KF/CAPE-PR substantially improves the model performance in reproducing lightning spatial patterns.

To evaluate the agreement quantitatively, we regress the WRF daily regional average flash densities against those measured by ENTLN. The daily regional averaged flash density is calculated by summing the total flash rates and dividing them by the corresponding regional size. The regressions are shown in Fig 1 (e) and (f); the correlation statistics are shown in Table 1. The regressions by forcing intercept equals to zero are also tested, and the results are unaffected.

Both models using the KF/CTH and KF/CAPE-PR parameterizations improve the correlation between modeled and observed lightning flash densities over the US domain. In the southeastern US, changing from G3 to KF convective scheme substantially increases the $R^2$ from 0.30 to 0.67 and reduces the slope from 2.08 to 0.94. Switching from CTH to CAPE-PR lightning parameterization only contributes a slight increment on the correlation. While the slopes close to unity both for KF/CTH and KF/CAPE-PR, we note that the improved scaling of the slope in KF/CAPE-PR is mainly caused by the scaling

factor of 0.5 applied to the southeast region. In this simulation, a constant linear coefficient for CAPE-PR is not adequate

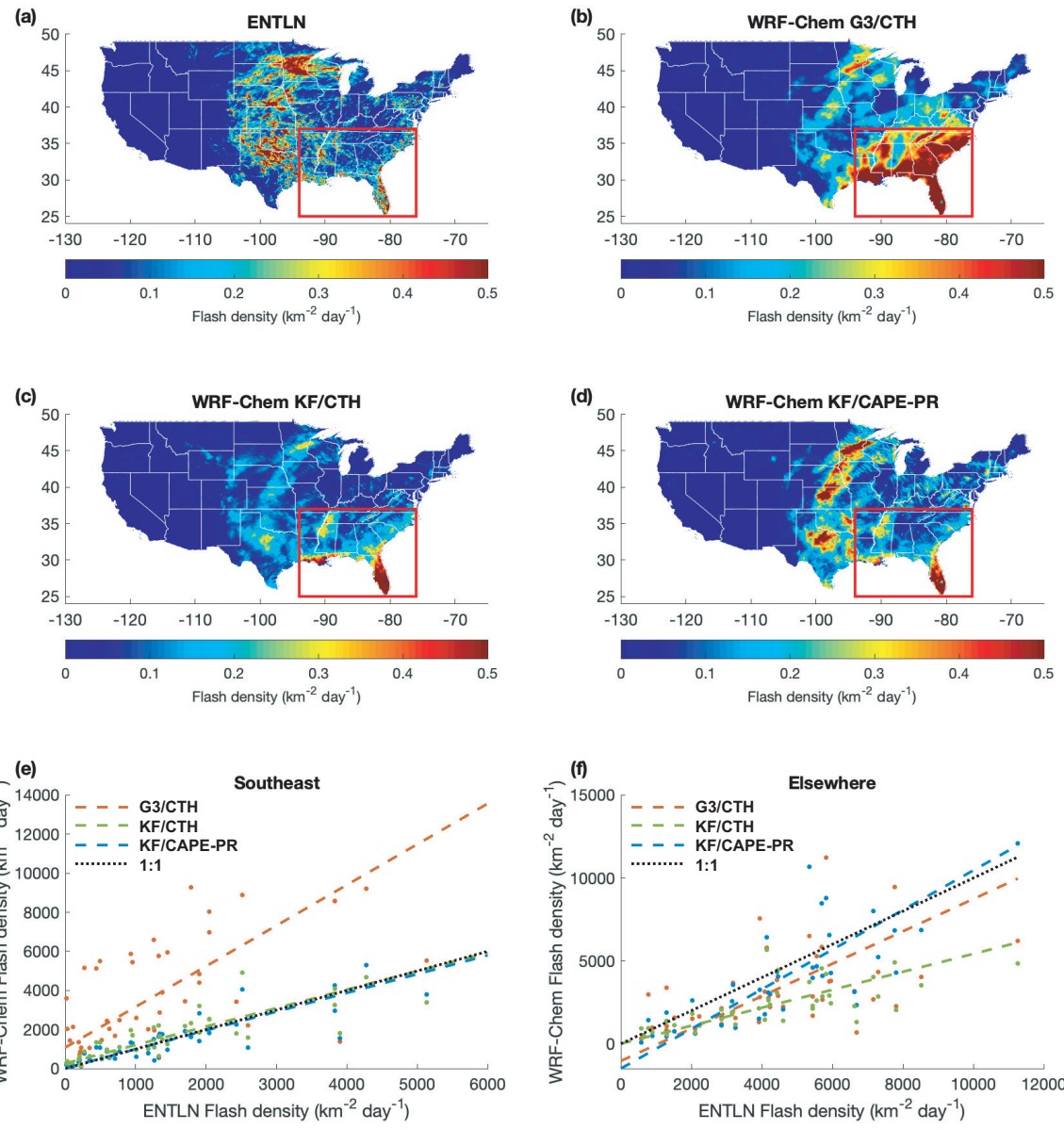

**Figure 1.** Observed flash densities from the ENTLN dataset (**a**) and WRF-Chem using three coupled convective-lightning parameterizations, the G3/CTH parameterization (**b**), the KF/CTH parameterization (**c**) and the KF/CAPE-PR parameterization (**d**), respectively. The correlation of total flash density per day between WRF-Chem outputs and ENTLN for the southeastern US (denoted by the red box in **a-d**) is shown in panel (**e**) and the correlation for elsewhere in CONUS is shown in (**f**). The model using G3/CTH is in red, KF/CTH is in green, and KF/CAPE-PR is in blue. Dash lines are corresponding fits. For slope and $R^2$, see Table 1.

|  |  | AMF G3/CTH | AMF KF/CAPE-PR | %ΔAMF | VCD G3/CTH | VCD KF/CAPE-PR | %ΔVCD |
|---|---|---|---|---|---|---|---|
| Sep 10 | Urban | 1.64 | 0.72 | -56.0 | $2.19\times10^{15}$ | $5.16\times10^{15}$ | 134.9 |
|  | Rural | 1.96 | 1.33 | -32.0 | $1.11\times10^{15}$ | $1.63\times10^{15}$ | 44.9 |
| Aug 24 | Urban | 1.07 | 0.95 | -11.3 | $2.56\times10^{15}$ | $2.64\times10^{15}$ | 3.1 |
|  | Rural | 1.23 | 1.25 | 1.60 | $1.91\times10^{15}$ | $1.82\times10^{15}$ | -4.6 |

**Table 2.** Differences for BEHR AMFs and tropospheric VCDs when using the a priori $NO_2$ profiles from models with CTH vs CAPE-PR parameterizations in the AMF calculation. For definitions of "urban" and "rural", see the text.

to represent the observed lightning over CONUS, in contrast to the finding of Romps et al. (2014). Elsewhere in CONUS, both the changes in convective scheme and lightning parameterization yield a better representation of lightning flash densities compared to the observation. The $R^2$ for KF/CAPE-PR improves significantly to 0.62 compared to both G3/CTH and KF/CTH. The slope for KF/CAPE-PR is 1.19, which is within the uncertainty of the detection efficiency of ENTLN. In general the KF/CAPE-PR lightning parameterization captures the day-to-day variation in flash densities better than the G3/CTH and KF/CTH parameterizations as shown by the improved $R^2$ values.

### 3.2 Comparison with observed vertical profiles

We compare the WRF $NO_2$ profile to the average vertical profile of $NO_2$ measured during DC3 and SEAC4RS in Fig 2. Data points are matched in time and space by finding the WRF-Chem output nearest in time and closest in space to a given observation. We only compare $NO_2$ profiles from WRF-Chem using KF/CAPE-PR against the one using G3/CTH.

The effect of lightning $NO_x$ on the profiles is indistinguishable close to the surface. In the upper and middle troposphere, both model simulations yields similar $NO_2$ vertical profiles compared to the measurements from DC3. WRF-Chem using KF/CAPE-PR performs slightly better between 200 hPa to 400 hPa but the negative bias still exists. $NO_x$ from both the observations and the models are very small in the middle troposphere between 400 hPa to 700 hPa.

Laughner et al. (2019) previously identified a high bias of WRF-Chem UT $NO_2$ versus SEAC4RS in the southeast US when using the G3/CTH parameterization. The model using the KF/CAPE-PR parameterization reduces this high bias of $NO_2$ in the middle and upper troposphere. The KF/CAPE-PR parameterization slightly overestimates $NO_2$ in the middle troposphere (400 - 530 hPa) and underestimates it in the upper troposphere ($< 280$ hPa), which is consistent with the comparison to observations from DC3 campaign.

### 3.3 Impact on BEHR $NO_2$ retrievals

In space-based retrievals of $NO_2$, the AMF is required to convert the slant column density (SCD) obtained by fitting the observed radiances into a vertical column density (VCD). The AMF depends on scattering weights (which describe the sensitivity of the measurement to different levels of the atmosphere) and an $NO_2$ profile which is either measured or simulated by a chemical transport model, such as WRF-chem. Over a dark surface, the scattering weights in the UT are up to 10x greater

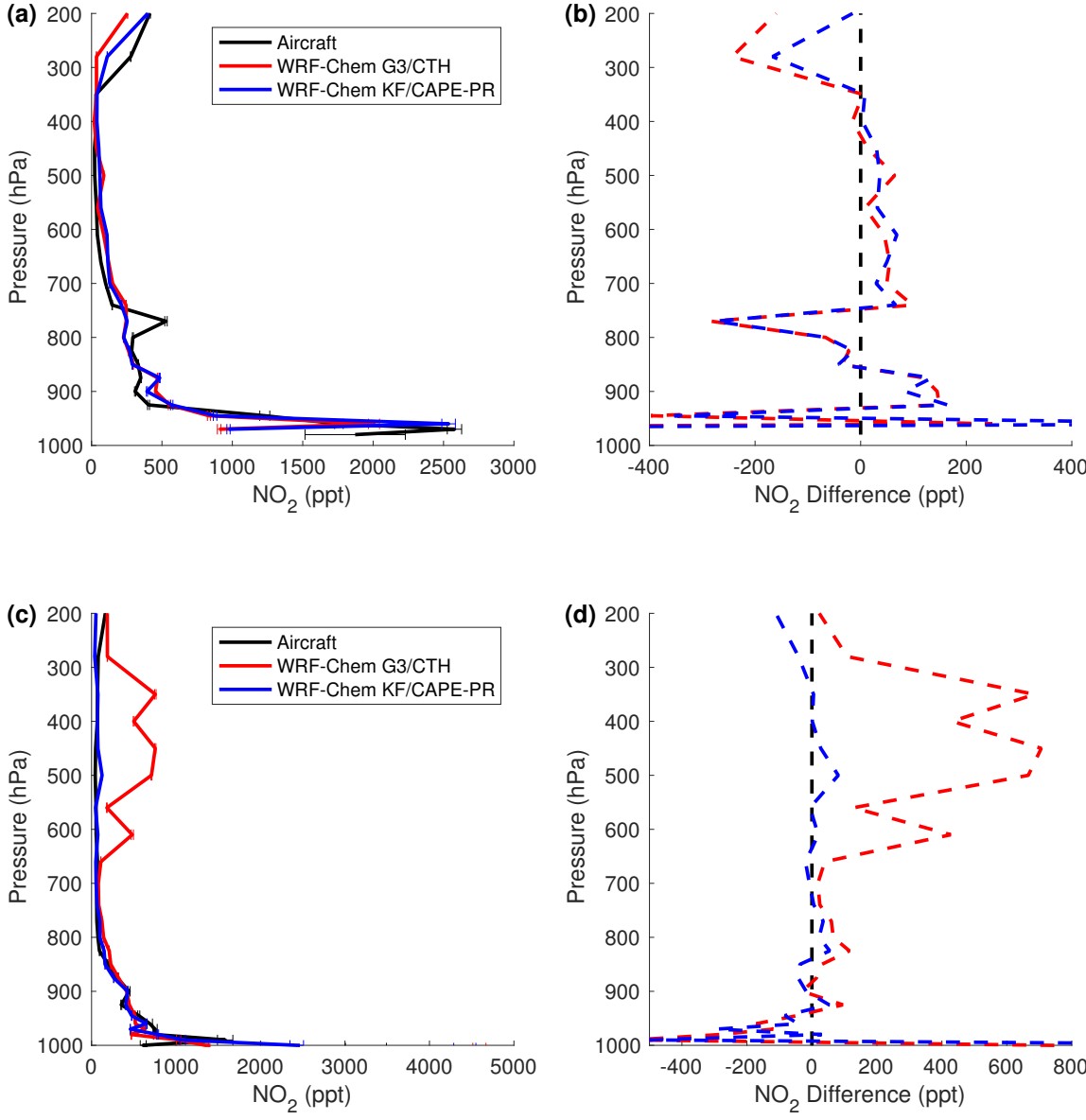

**Figure 2.** Comparison of WRF-Chem and aircraft NO₂ profiles from the **(a,b)** DC3, **(c,d)** SEAC4RS campaigns. Vertical NO₂ profiles are shown in **(a,c)**, the solid line is the mean of all profiles and the bars are 1 standard deviation for each binned level. The corresponding absolute difference compared to observations are shown in **(b,d)**. Aircraft measurements are shown in black, WRF-Chem using G3/CTH parameterization in red and WRF-Chem using KF/CAPE-PR parameterization in blue.

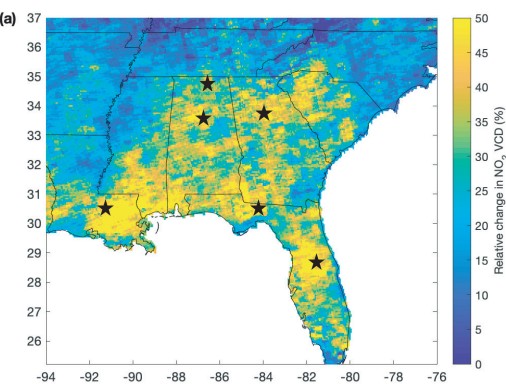 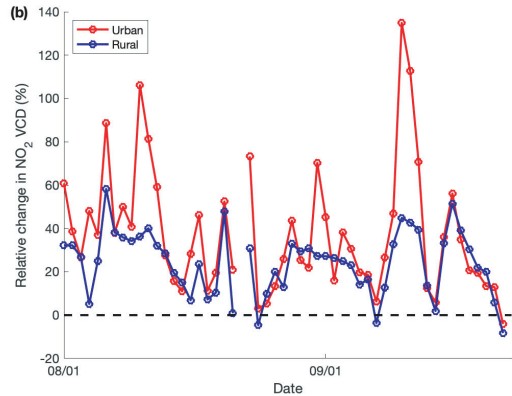

**Figure 3.** Relative change in BEHR $NO_2$ VCD over the southeastern US switching the source of a prior $NO_2$ profiles from WRF-chem outputs using G3/CTH to one using KF/CAPE-PR lightning parameterization. **(a)** shows the mean spatial distribution of the changes from Aug 01 to Sep 23, 2013 and **(b)** shows the temporal variation over urban and rural areas. Only observations with cloud fraction less than 20% are included. Medium to large cities, including Atlanta, GA; Huntsville, AL; Birmingham, AL; Tallahassee, FL; Orlando, FL; and Baton Rouge, LA, are marked by stars in panel **(a)**.

than near the surface, due to the greater probability that a photon that reaches the lower troposphere will be absorbed by the surface. Therefore, errors in the UT $NO_2$ profile can have large effects on the AMF (e.g. Laughner and Cohen, 2017). Here, we investigate how the $NO_2$ profiles simulated by the KF/CAPE-PR parameterization affect the BEHR $NO_2$ retrievals.

Fig. 3(a) shows the relative change in tropospheric VCD averaged between Aug 01 to Sep 23, 2013 induced by changing the a priori profiles from the model using G3/CTH to the one using the KF/CAPE-PR lightning parameterization. The relative enhancement of VCD is 19% on average over southeast US but it varies significantly.

We follow the same algorithm used in Laughner and Cohen (2017) to determine if the result is significant. The overall uncertainty due to AMF calculation for BEHR v3.0B is smaller than 30% during the study period (Sec 6 in supplementary from Laughner et al. (2019)). Over 90% of the uncertainty attributes to the a prior $NO_2$ profiles, the tropopause and cloud pressures. As each grid in Fig. 3(a) is the average of 45±9 pixels, the reduced uncertainty is less than 4.5%. The overall change in VCD is four times larger than the reduced uncertainty. The switch of lightning parameterization leads to changes in VCD exceeding the averaged uncertainty in ~94% of pixels in the southeast region of US.

The spatial pattern in Fig. 3(a) suggests that the magnitude of the improved representation of lightning is quite different in urban and rural areas. The cities indicated by stars and their vicinity regions are associated with substantial increase in $NO_2$ VCD. To quantify this, we define urban and rural areas by difference in column $NO_2$ calculated from WRF-Chem without $LNO_x$. Urban ares are the top 5% of columns with the average VCD of $2.2 \times 10^{15}$ mole $cm^{-2}$. The selected rural areas have the same size as urban areas and the average VCD is $0.72 \times 10^{15}$ mole $cm^{-2}$. Fig 3(b) shows the relative change in VCD over the urban and rural areas as a function of time. The increase in VCD due to the change in profiles is more pronounced over urban areas with averaged relative change of ~38% compared to the average change of ~24% in rural areas. Changes in

urban VCDs span -10% to 135%. In contrast, using the $NO_2$ profiles produced by the KF/CAPE-PR simulation leads to only maximum 58.3% increase in VCD over rural areas.

Table 2 presents the AMF and VCD obtained from using a priori profiles with G3/CTH or KF/CAPE-PR lightning parameterizations as well as the relative changes on Sep 10 and Aug 24, 2013. The corresponding a priori $NO_2$ profiles and scattering weights over urban and rural areas are shown in Fig. S3. The G3/CTH parameterization has substantially more lightning than observed and thus places a large fraction of $NO_2$ in the upper troposphere whereas the KF/CAPE-PR has less lightning and is more consistent with observations. The resulting profiles of modeled $NO_2$ are more dominated by boundary layer $NO_2$ and less sensitive to lightning. Sep 10 is an example of one day when the change in $NO_2$ profiles has a very large impact on the $NO_2$ VCDs. The WRF-Chem using G3/CTH parameterization places a large amount of $NO_2$ between 200-600 hPa with the maximum value comparable to the near surface $NO_2$ over the urban areas. The calculated AMF is predominantly determined by lightning $NO_2$ due to the combination of higher scattering weight and larger $NO_2$ in the middle and upper troposphere. The change in AMF is -56.0% over urban areas and -32.0% over rural areas; the corresponding VCD increases by 134.9% and 44.9%, respectively. In contrast, Aug 24 is an example where the lightning parameterization has very little effect. While the positive bias in $NO_2$ aloft is also observed by using G3/CTH parameterization, the amount of $NO_2$ in the middle and upper troposphere is smaller than Sep 10. It leads to lower sensitivity in AMF to the erroneous $NO_2$ caused by the lightning parameterization. With smaller relative change in AMF, the relative change in VCD is 3.1% over urban areas and -4.6% over rural areas.

## 4 Discussion

Here, we apply the improved KF/CAPE-PR simulation to the problem of constraining $LNO_x$ production over CONUS. To do so, we vary the lightning $NO_x$ production rate prescribed in WRF-Chem to produce the simulated map of $NO_2$ VCD, and compare against OMI $NO_2$ retrievals using a priori profiles from model simulations with the same $LNO_x$ production rate. In our model-satellite comparisons the averaging kernel is applied to remove the representative errors introduced by a priori knowledges of $NO_2$ vertical profiles (Boersma et al., 2016). Figure 4 shows the difference between satellite retrieved $NO_2$ VCD and model simulated $NO_2$ VCD without lightning $NO_x$ **(a)** and with lightning $NO_x$ production rate of 500 mol NO flash$^{-1}$ **(b)** averaged between May 13 to June 23, 2012. Figure S4 shows difference plots with varied lightning $NO_x$ production rates (400 and 665 mol NO flash$^{-1}$). The corresponding root-mean-square errors (RMSE) are included in Table S1. $LNO_x$ production rate of 500 mol NO flash$^{-1}$ yields the lowest RMSE of $0.41 \times 10^{15}$ mole cm$^{-2}$ between modeled and observed $NO_2$ VCD over CONUS. This is at the high end of previous estimates of the lightning $NO_x$ production rate (16-700 mol NO flash$^{-1}$).

The RMSE for urban areas (top 5% of $NO_2$ VCD simulated by WRF-Chem without $LNO_x$) remains at high value (~0.9-1.3$\times 10^{15}$ mole cm$^{-2}$) when switching the $LNO_x$ production rate. It indicates that the bias in the modeled VCD over urban areas is more likely due to surface $NO_2$. The RMSE for non-urban areas shows pronounced change with varied $LNO_x$ production rate. Excluding urban areas lowers the RMSE to $0.37 \times 10^{15}$ mole cm$^{-2}$ for $LNO_x$ production rate of 500 mol NO flash$^{-1}$. The RMSEs are significant considering the uncertainty for retrievals. During the average time period, $32 \pm 6$ pixels

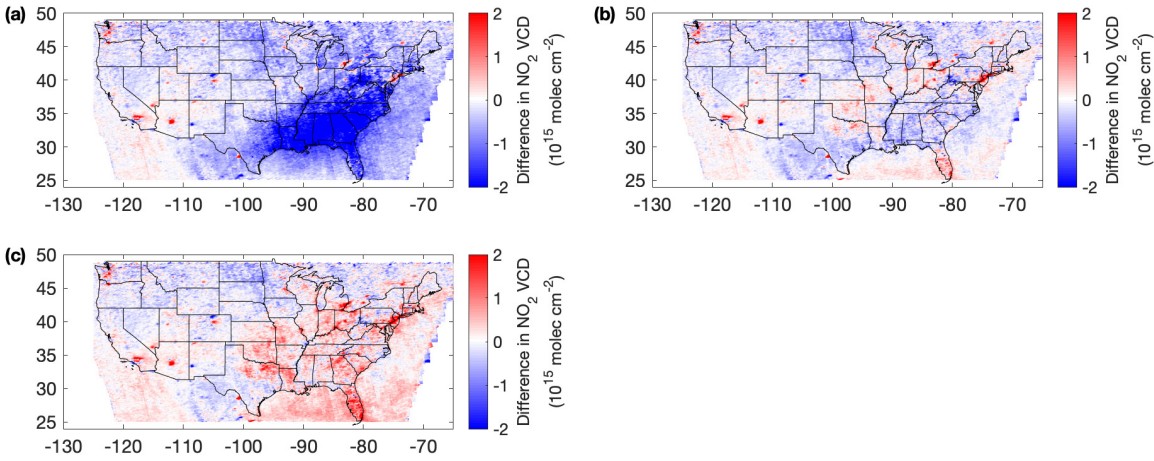

**Figure 4.** Difference in $NO_2$ VCD between BEHR retrievals and WRF-Chem ("WRF-Chem" − "BEHR"). **(a)** excludes $LNO_x$ in model simulation, **(b)** adds $LNO_x$ emission with production rate of 500 mol NO flash$^{-1}$. **(c)** includes the same $LNO_x$ emission as **(b)** but uses $NO_2$ profiles scaled upward by 60% at pressure lower than 400 hPa. The average time covers May 13 to June 23, 2012. Pixels with cloud fraction larger than 0.2 are filtered out in the analysis.

contribute to each value in the plots. While the global mean uncertainty for tropospheric $NO_2$ VCD retrievals is $1\times10^{15}$ mole cm$^{-2}$ (Bucsela et al., 2013), the reduced uncertainty in our analysis is ~$0.2\times10^{15}$ mole cm$^{-2}$. The calculated RMSEs are twice of the uncertainty.

However, we note that this lightning $NO_x$ estimate is systematically biased high due to the negative bias in $[NO_2]/[NO_x]$
5  ratio in the middle and upper troposphere. The satellite observed $NO_2$ column serves as a proxy for total $NO_x$ emitted by lightning. The rapid interconversion between NO and $NO_2$ reaches the photochemical steady state in a short time (~120s). Consequently, if the model kinetics result in an incorrect NO-$NO_2$ photochemical steady state ratio, this error will propagate into the $LNO_x$ production estimate. Comparisons against aircraft measurements show $[NO_2]/[NO_x]$ ratio in the WRF-Chem simulations is around 40% smaller than observations in upper troposphere (Fig. S5). Given that the simulated $[NO_2]/[NO_x]$
10  is too small, the model will simulate smaller $NO_2$ VCDs per unit of $LNO_x$ emitted, requiring a greater $LNO_x$ production efficiency to match satellite $NO_2$ VCD observations. Comparison of modeled $NO_2$ columns recalculated with $NO_2$ profiles scaled up by 60% (the ratio of observed and modeled $[NO_2]/[NO_x]$) at pressure levels where $p < 400$ hPa and observations is shown in Fig. 4 **(c)**. This suggests that the 500 mol NO flash$^{-1}$ is greater than the actual $LNO_x$ production rate when the bias caused by $[NO_2]/[NO_x]$ ratio is accounted for.

15  Several recent studies also report an underestimate in modeled $[NO_2]/[NO_x]$ ratios in SE US(Travis et al., 2016; Silvern et al., 2018); both feature observations from SEAC4RS field campaign to validate model simulations. Silvern et al. (2018) suggests the underestimate is either caused by an unknown labile $NO_x$ reservoir species or error in reaction rate constant for the NO+$O_3$ reaction and $NO_2$ photolysis reaction. In contrast, Nault et al. (2017) utilizes measurements from DC3 field

campaign and demonstrates a positive bias in modeled $[NO_2]/[NO_x]$ ratio compared against observations. Understanding the difference in $[NO_2]/[NO_x]$ between model and observations requires additional study, but is crucial to reducing the uncertainty in $LNO_x$ estimates.

## 5  Conclusions

We implement an alternative lightning parameterization based on convective available potential energy and precipitation rate into WRF-Chem and couple it with Kain Frisch convective scheme. We first validate it by comparing against lightning observations and find that the model reproduces day-to-day variation of lightning flashes in the southeastern US after the switch of convective scheme and the switch of lightning parameterization contributes to the improvement of the lightning representation elsewhere in the US. We also compare the simulated $NO_2$ profiles against aircraft measurements and find that the simulated

$NO_2$ using KF/CAPE-PR is more consistent with observations in the mid and upper troposphere.

The improved lightning $NO_2$ simulation has significant impact on AMFs and VCD of $NO_2$. Over the southeastern US the AMF is reduced by 16% on average leading to a 19% increase in the $NO_2$ VCD. The effects on AMF and on VCD are very locally dependent. The VCD increase over urban areas is more pronounced and can be up to over 100%. This study indicates that the erroneous representation of lightning $NO_2$ in a priori profiles is an important source of bias for satellite retrievals. The

model-satellite $NO_2$ column comparison suggests 500 mol NO flash$^{-1}$ is the upper bound for the estimate of lightning $NO_x$ production rate.

*Data availability.*  The experimental branch of BEHR v3.0B product used in this study is hosted by UC Dash (Zhu et al., 2019a, b) as well as on behr.cchem.berkeley.edu. The BEHR algorithm is available at https://github.com/CohenBerkeleyLab/BEHR-core/ (Laughner and Zhu, 2018). The revised WRF-Chem code is available at https://github.com/CohenBerkeleyLab/WRF-Chem-R2SMH/tree/lightning (Zhu

and Laughner, 2019).

*Author contributions.*  RCC directed the research and QZ, JLL and RCC designed this study; JLL and QZ developed BEHR products; QZ performed the analysis and prepared the manuscript with contributions from JLL and RCC. All authors have reviewed and edited the paper.

*Competing interests.*  The authors declare no competing interests.

*Acknowledgements.*  This work was supported by a NASA ESS Fellowship NNX14AK89H (Laughner), NASA grants NNX15AE37G and

80NSSC18K0624, and the TEMPO project SV3-83019. We acknowledge use of the Savio computational cluster resource provided by the Berkeley Research Computing program at UC Berkeley which is supported by the UC Berkeley Chancellor, Vice Chancellor for Research,

and Chief Information Officer. We thank Earth Networks Company for providing the Earth Networks Total Lightning Network (ENTLN) datasets. We appreciate use of the WRF-Chem preprocessor tool (mozbc) provided by the Atmospheric Chemistry Observations and Modeling Lab (ACOM) of NCAR and use of MOZART-4 global model output available at http://www.acom.ucar.edu/wrfchem/mozart.shtml.

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
