# Peer review of "Lightning NO2 simulation over the Contiguous US and its effects on satellite NO2 retrievals"

_Atmospheric Chemistry and Physics, 2019_

## Referee Comment (RC1) · Anonymous Referee #1 · 3 May 2019

The Zhu et al. manuscript presents results of WRF-Chem simulations conducted for the DC3 and SEAC4RS experimental periods with two convective parameterizations, and lightning flash rates were predicted using a different scheme associated with each convective scheme. Lightning NOx (LNOx) was predicted and compared with a version of the BEHR OMI NO2 retrievals and with aircraft data from the two experiments. Results show improved prediction of flash rates with the Grell 3D convection and CAPE-PR lightning scheme compared with Kain-Fritsch convection and cloud-top-height lightning scheme. LNOx production of 500 moles per flash produces the best model comparison with the OMI retrievals, but the model produces an NO2/NOx ratio in the middle and upper troposphere that is smaller than observed by aircraft, suggesting that the production is really less than 500 moles/flash.

[Figure]

Here is my major issue with this manuscript: Throughout the manuscript the authors state that the lightning prediction is improved with use of CAPE-PR compared with CTH. However, the lightning schemes are run with different convective parameterizations which are going to produce different convective characteristics (locations, timing, frequency, amounts of precipitation, etc.). Therefore, they would need to run CAPE-PR with Kain Fritsch to truly be able to say that CAPE-PR is better. If making this additional model run is not possible, I would then suggest that throughout the paper the authors refer to CAPE-PR as Grell3D/CAPE-PR and refer to CTH as KF/CTH to reflect the fact that it is a combination of convection and lightning schemes that are producing the difference they see in lightning flash rates.

Specific comments: p. 2, line 11: give examples of near-filed analyses (e.g., Huntrieser et al. (several papers); Pollack et al., 2016). add another sentence: Near-field estimates of LNOx per flash have also been make through use of cloud-resolved models with LNOx production constrained by observed flashes and aircraft data from storm anvils (e.g., DeCaria et al., 2005; Ott et al., 2010; Cummings et al., 2013).

p. 3, line 4: flash count frequency distribution over time,....

p. 3, ine 9: need a reference for CAPE-PR here

p. 3, line 15: Provide the time periods that are being simulated here. The reader needs to know if 2012 emissions are appropriate. Otherwise, the reader doesn't learn the simulating periods until Section 2.3.

p. 4, line 16: the neutral buoyancy level

p. 4, lines 14-22: Won't the difference in flash rate between the two model runs be partially due to different convective parameterizations and partially due to different flash rates schemes? Here is where the authors either need to add another model run (KF with CAPE-PR) or start calling the two runs Grell3D/CAPE-PR and KF/CTH.

p. 4, line 29: was this detection efficiency value applied to teh ENTLN data? Flash

counts should be divided by 0.7.

p. 5, line 12: should the reference by 2019 instead of 2018? 2019 is the one with v3.0B in the title.

p. 5, line 15: what is "NASA tropopause temperature"? Is it from the MERRA-2 product?

p. 6, lines 14-15: But, how much of this improvement might be due to use of KF convection rather than Grell 3D convection? I don't think you can conclude that one lightning scheme is better than the other with these two simulations that use different convection schemes.

p. 8, lines 4-5: Need to point out that this is really only true for the CTH model runs for SEAC4RS. Both model and observations are very small in this layer for DC3.

p. 11, lines 4-5: Here again, this conclusion needs to be modified. See above.

p. 15, lines 1-2: Update Laughner and Cohen reference

Figure S1: Since LIS only observes a swath across this region for a few minutes each day, I assume that the ENTLN data are subsetted in time to match the LIS overpass times. Is this correct? If so, you need to say that in the caption. Or, if the ENTLN data are really for the entire day/night for all days, then the comparison is not valid. The c) panel of the figure does not look to be correct. Is it really ENTLN - LIS rather than LIS - ENTLN? I see some pixels where neither appears to be correct.

---

## Referee Comment (RC2) · Anonymous Referee #2 · 12 May 2019

A new lightning NOx parameterization was developed, and the model simulations were applied to compare to in situ NO2 observations during DC3 and SEAC4RS. The implications for OMI NO2 VCD retrievals were also analyzed. While I think the paper covers interesting areas of research, the results from this paper are in line with previous studies. It would be a stronger paper if more significant improvements in either modeling or analysis results were made.

Some of the discussion is rather odd. A few of the issues were raised in my initial review but no modifications were made to the paper. These issues are not complex but do require more in-depth thinking than what was given in the manuscript. I think that the uncertainties in lightning measurements and modeling, satellite measurements and retrievals, and in situ measurements should be clearly acknowledged.

[Figure]

(1) Lightning modeling uncertainty

I pointed out in the initial review that the abstract statement in line 11-12 on page 1 should be deleted. Comparing NO2 VCDs between two parameterizations is not meaningful because the amount of lightning NOx (LNOx) is a function of specified IC/CG ratio, NOx yield per flash, and the vertical distribution of LNOx. Values can be easily modified to produce similar LNOx values between the two parameterizations. For the same reason, line 12-13 on page 11 in the conclusion section should be deleted. The NOx production rate per (IC or CG) flash cannot be determined with available observations.

The abstract statement in line 5-6 on page 1 is based on model comparison with SEAC4RS data, when the lightning activity in the Southeast is relatively low. Unless it is a science objective, aircraft in missions like SEAC4RS usually flies in sunny days and steers away from thunderstorms. The 50% reduction is also only specific to the CAPE-PR parameterization in this paper and it offers little value to other lightning parameterizations.

It is not surprising that the CTH based parameterization doesn't work well. It's a poor choice in WRF-Chem, but I presume that it's a good reason to compare it to a new scheme. It would be better that other parameters, say those used by Allen et al. (2002) and more recently Luo et al. (2017), were used.

A third issue raised in the initial review is line 17-19 on page 4. Which convection scheme was used for the results presented later in the paper? Are there differences? I did not find the results comparing the two convective schemes. Zhao et al. (2009) compared MM5 Grell and WRF KF schemes and found large differences. My understanding is that the Grell scheme in WRF does not have the large bias in MM5. It should be discussed.

(2) Lighting measurement uncertainty

[Figure]

**ACPD**

Section 2.2 described the ENTLN lightning data and its use in this paper. Figure 1 in the paper by Luo et al. (2017) compared the lightning distributions observed by ENTLN to NLDN. For the data they used, ENTLN had more IC and CG lightning flash rates than NLDN. The lower IC flash rates in NLDN are likely due to a lower detection efficiency. However, the CG flash rates in NLDN are also lower. As implied by Luo et al. (2017), the NLDN CG flash rate data have been the "gold standard" in previous LNOx studies over the US. The distributions of NLDN flash rates were also different from ENTLN data in that work. The uncertainties of ENTLN data should be acknowledged. It is another reason the two statements on lightning parametrization in the abstract (discussed above) are not robust science results and should be taken out. Some explanation is due for the reasons of not using NLDN data.

The comparison in Figure S1 is inadequate since it is only for one day only. LIS observations are mostly for IC flash rates and there are good reasons to believe that a CG flash produces more NOx than an IC flash (see the relevant discussion in Luo et al. (2017)).

(3) OMI retrieval uncertainties

The comparisons of Figures 4 and S2 can only be used as a qualitative not quantitative measure of lightning NOx in the model. For clean regions of the SE, the difference is on the order of $1 \times 10^{15}$ molec cm-2. Considering that the OMI uncertainty is larger than $1.5 \times 10^{15}$ molec cm-2, it is difficult to say which sensitivity simulation is quantitatively better. There are additional uncertainties in OMI retrievals including surface albedo, cloud, and background noise. With the uncertainties in mind, the difference among the lightning sensitivity simulations in Figure S3 is therefore insignificant. Appropriate discussion on the uncertainties of OMI retrievals and the implications for this paper should be included.

There is no description on how OMI data under high cloud-fraction conditions are treated. Those data cannot be used in the comparisons of Figures 4 and S2.

[Figure]

(4) Uncertainty of the upper tropospheric NO2/NOx ratio

In sections 4, the uncertainty of the upper tropospheric NO2/NOx ratio was discussed. This issue doesn't affect model comparison with in situ observations when NO measurements are available. Comparisons with in situ NO, O3, and JNO2 should be included with the discussion of Figure 2. To evaluate model lightning NOx simulations, using NO measurements will get around the uncertainty of NO2/NOx ratio. Therefore, the last statement of the conclusion section (line 13-14 on page 11) is inappropriate and should be removed.

This uncertainty affects the retrieval of OMI data only if the unknown interferences by other nitrates are insignificant. Furthermore, the uncertainties of OMI data may mask out the effects. More detailed discussion should be included.

Other comments:

(1) P. 4, Line 26-27, Eq. (2), Luo et al. (2017) used a formula of , what is the reason for not including the other terms? What are the reasons for not using UMF or CPR?

(2) P. 8, Figure 3, Is the large urban VCD change mostly due to Orlando? Figure 1 shows very little lightning activity in the other SE region.

(3) P. 9, Line 6-7, the 19% value should be compared to Zhao et al. (2009).

(4) Figures are hard to read in general.

---

## Author Comment (AC1) · 15 Jul 2019

**Lightning NO$_2$ simulation over the Contiguous US and its effects on satellite NO$_2$ retrievals**
**Response to Anonymous Referee #1**

Qindan Zhu, Joshua L. Laughner and Ronald C. Cohen

July 12, 2019

We thank the reviewers for their positive response and very careful reading of both the main article and the supplement. Our responses to the reviewer's comments and detailed changes made to the manuscript are addressed below. The reviewer's comments will be shown in red, our response in blue, and changes made to the paper are shown in black block quotes. Unless otherwise indicated, page and line numbers correspond to the original paper. Figures, tables, or equations referenced as "R$n$" are numbered within this response; if these are used in the changes to the paper, they will be replaced with the proper number in the final paper. Figures, tables, and equations numbered normally refer to the numbers in the original discussion paper.

Here is my major issue with this manuscript: Throughout the manuscript the authors state that the lightning prediction is improved with use of CAPE-PR compared with CTH. However, the lightning schemes are run with different convective parameterizations which are going to produce different convective characteristics (locations, timing, frequency, amounts of precipitation, etc.). Therefore, they would need to run CAPE-PR with Kain Fritsch to truly be able to say that CAPE-PR is better. If making this additional model run is not possible, I would then suggest that throughout the paper the authors refer to CAPE-PR as Grell3D/CAPE-PR and refer to CTH as KF/CTH to reflect the fact that it is a combination of convection and lightning schemes that are producing the difference they see in lightning flash rates.

We appreciate the reviewer pointing out the inconsistency in the convective schemes. To be clarified, we used two model runs in the manuscript, one uses Grell 3D convective scheme with CTH lightning parameterization ("G3/CTH"), and another one uses Kain Fritsch convective scheme with CAPE-PR lightning parameterization ("KF/CAPE-PR"). The first one is the only option provided by WRF to parameterize lightning at convective parameterized scale.

We agree that the switch of convective scheme will affect the parameterized lightning flashes and as a result, we can't conclude the improvement in lightning flash representation in WRF-Chem is solely due to the new implemented CAPE-PR lightning parameterization. Therefore, we made another WRF-Chem run with Kain Frisch convective scheme and CTH lighting parameterization, referred as "KF/CTH". We add the description of this model run in Section 2.1.

"We analyze WRF-Chem outputs from three model runs. The first run, **referred as "G3/CTH"**, is consistent with Laughner and Cohen (2017); it selects the Grell 3D ensemble cumulus convective scheme (Grell, 1993; Grell and Dvnyi, 2002) and the CTH lightning parameterization. The Grell 3D convective scheme readily computes the neutral buoyancy level which serves as the optimal proxy for cloud top height (Wong et al., 2013). **The "G3/CTH" is the only option for the coupled convective-lighting parameterization used in WRF-Chem at a non-cloud resolving resolution (12 km). In addition, we run WRF-Chem with the CTH lightning parameterization coupled with the Kain-Fritsch cumulus convective scheme (Kain and Fritsch, 1990; Kain, 2004) ("KF/CTH") to test the effect of switching convective schemes. In the "KF/CTH" parameterization, the cloud top height is the level where the updraft vertical velocity equals to zero.** Another run, referred as "KF/CAPE-PR"**,** selects the Kain-Fritsch cumulus convective scheme and the CAPE-PR lightning parameterization described above..."

In the context, we change CTH to either "G3/CTH" or "KF/CTH" and change CAPE-PR to "KF/CAPE-PR" accordingly. In Section 3.1, we added comparison of "KF/CTH" parameterized lightning flashes against ENTLN both in the text, Fig 1 and Table 1 (also labeled as Fig. R1 and Table R1 in this response).

[revised manuscript text omitted]

p. 3, line 15: Provide the time periods that are being simulated here. The reader needs to know if 2012 emissions are appropriate. Otherwise, the reader doesn?t learn the simulating periods until Section 2.3.
We added the study time period at the beginning of Section 2.1

"This study applies the Weather Research and Forecast Model coupled with Chemistry (WRF-Chem) version 3.5.1 **to the time periods May to June, 2012 and August to September, 2013.**"

p. 4, line 16: the neutral buoyancy level
Corrected, thanks.

p. 4, lines 14-22: Won't the difference in flash rate between the two model runs be partially due to different convective parameterizations and partially due to different flash rates schemes? Here is where the authors either need to add another model run (KF with CAPE-PR) or start calling the two runs Grell3D/CAPE-PR and KF/CTH.
We added another model run referred as KF/CTH. The revisions are summarized above.

p. 4, line 29: was this detection efficiency value applied to the ENTLN data? Flash counts should be divided by 0.7.
We did a more thorough survey on the detection efficiency of ENTLN and decided to correct it to 88%. As the detection efficiency of ENTLN varies by time, region, lightning type and the reference datasets, the choice of 88% is based on following elements:

1. Local studies comparing ENTLN to rocket-triggered lightning data at Florida report detection efficiency of ENTLN to be 89% during 2009-2012 (Mallick et al., 2015) and 99% during 2014-2015 (Zhu et al., 2017).

2. Lapierre et al. (submitted) found out the average detection efficiency for flashes observed by ENTLN+NLDN was 88% over CONUS relative to space-based TRMM-LIS during May-August, 2014.

In the context, we decide not to correct the ENTLN flash counts using the detection efficiency (88%) referring to Fig S1. In Fig S1, we matched the ENTLN data to LIS flashes both in time and space after the correction of LIS data based on its detection efficiency (Cecil et al., 2014) during the study period May 13-June 23, 2012. Both two datasets are then summed on 0.5°x0.5° grid cells and they show median correlation with slope of 1.0. While it indicates uncorrected ENTLN during study time period shows the best agreement with LIS observation, the detection efficiency of ENTLN is only considered as a source of uncertainty when comparing modeled lightning flashes against ENTLN. We added the following text in Section 2.2:

> "Compared to National Lightning Detection Network (NLDN), ENTLN is selected for high detection efficiencies of both CG and IC flashes. The average detection efficiency for total flashes observed by ENTLN was 88% over CONUS relative to the space-based Tropical Rainfall Measurement Mission (TRMM) Lightning Imaging Sensor (LIS) (Lapierre et al. (submitted), private communication). Shown in Fig. S1, we matched the ENTLN data to LIS flashes both in time and space after the correction of LIS data based on its detection efficiency (Cecil et al., 2014) during May 13-June 23, 2012. It shows a median correlation ($R^2 = 0.51$) with the slope of 1.0, indicating the ENTLN data during the study time period is in agreement with the LIS observation. We use the ENTLN for analysis as reported and consider the detection efficiency of ENTLN as a source of uncertainty when comparing the modeled lightning flashes."

p. 5, line 12: should the reference by 2019 instead of 2018? 2019 is the one with v3.0B in the title.

Laughner et al. (2018) is the first paper describing the upgrade to BEHR v3.0B and Laughner et al. (2019) is the followup paper evaluating BEHR 3.0B. In the context Laughner et al. (2018) should be the one for citation

p. 5, line 15: what is "NASA tropopause temperature"? Is it from the MERRA-2 product? We reformatted the sentence for clarification:

> "First, Instead of calculation based on temperature profiles from WRF-Chem (Mak et al., 2018), the tropopause pressure is switched to **GEOS-5 monthly tropopause pressure which is consistent with NASA Standard Product (SP2).**"

p. 6, lines 14-15: But, how much of this improvement might be due to use of KF convection rather than Grell 3D convection? I don?t think you can conclude that one lightning scheme is better than the other with these two simulations that use different convection schemes.

We addressed this question in the response above. Please refer to our response to the first comment.

p. 8, lines 4-5: Need to point out that this is really only true for the CTH model runs for SEAC4RS. Both model and observations are very small in this layer for DC3.

We found using the percentage change in $NO_2$ profiles is misleading. The large bias in $NO_2$ comparing against DC3 observation between 400 to 700 hpa is due to the small values both in observation and simulation. To avoid this, we replace the percentage change with absolute difference in Fig. 2 (also labeled as Fig. R2 in this response).

In the context, we corrected the sentence to:

> "$NO_x$ from both the observations and the models are very small in the middle troposphere between 400 hPa to 700 hPa."

p. 11, lines 4-5: Here again, this conclusion needs to be modified. See above.
We modified the conclusion accordingly, please refer to the response to the first comment.

p. 15, lines 1-2: Update Laughner and Cohen reference
Corrected, thanks!

Figure S1: Since LIS only observes a swath across this region for a few minutes each day, I assume that the ENTLN data are subsetted in time to match the LIS overpass times. Is this correct? If so, you need to say that in the caption. Or, if the ENTLN data are really for the entire day/night for all days, then the comparison is not valid. The c) panel of the figure does not look to be correct. Is it really ENTLN - LIS rather than LIS - ENTLN? I see some pixels where neither appears to be correct.

We add a more detailed description of Fig. S1 in Section 2.2:

> "Shown in Fig. S1, we matched the ENTLN data to LIS flashes both in time and space after the correction of LIS data based on its detection efficiency (Cecil et al., 2014) during May 13-June 23, 2012. It shows a median correlation ($R^2 =$ 0.51) with the slope of 1.0, indicating the ENTLN data during the study time period is in agreement with the LIS observation...."

It's ENTLN-LIS. Thanks for pointing out the error, we update Fig. S1 (labeled as Fig. R3 in this response) and also modified the caption:

[Figure]

Figure R2: Comparison of WRF-Chem and aircraft $NO_2$ profiles from the **(a,b)** DC3, **(c,d)** SEAC4RS campaigns. Vertical $NO_2$ profiles are shown in **(a,c)**, the solid line is the mean of all profiles and the bars are 1 standard deviation for each binned level. The corresponding absolute difference compared to observations are shown in **(b,d)**. Aircraft measurements are shown in black, WRF-Chem using G3/CTH parameterization in red and WRF-Chem using KF/CAPE-PR parameterization in blue.

[Figure]

Figure R3: Comparison between flash rates observed by ENTLN and Lightning Imaging Sensor (LIS). **ENTLN data is matched to corrected LIS flashes both in time and space during May 13-June 23, 2012, and both datasets are summed onto 0.5°x 0.5°grid spacing.** (a,b) shows the spatial pattern of lightning flash rates measured by LIS (a) and ENTLN (b). The plot region covers 20°N - 38°N and 130°W - 65°W. (c,d) are corresponding absolute difference and scatter plots between LIS and ENTLN. **LIS data is corrected using the detection efficiency from** Cecil et al. (2014).


**Lightning NO$_2$ simulation over the Contiguous US and its effects on satellite NO$_2$ retrievals**

**Response to Anonymous Referee #2**

Qindan Zhu, Joshua L. Laughner and Ronald C. Cohen

July 12, 2019

We thank the reviewers for very careful reading of both the main article and the supplement. Our responses to the reviewer's comments and detailed changes made to the manuscript are addressed below. The reviewer's comments will be shown in red, our response in blue, and changes made to the paper are shown in black block quotes. Unless otherwise indicated, page and line numbers correspond to the original paper. Figures, tables, or equations referenced as "R$n$" are numbered within this response; if these are used in the changes to the paper, they will be replaced with the proper number in the final paper. Figures, tables, and equations numbered normally refer to the numbers in the original discussion paper.

While I think the paper covers interesting areas of research, the results from this paper are in line with previous studies. It would be a stronger paper if more significant improvements in either modeling or analysis results were made.

We appreciate the reviewer's positive comments on the overview of the paper. The significance of this study is to improve model performance in representing lightning and lightning NO$_x$ in WRF-Chem and to directly couple those components to a high spatial resolution OMI NO$_2$ retrieval.

Currently, except for convective resolved runs, lightning NO$_x$ scheme is disable by default in WRF-Chem, and the only option provided is combing CTH lightning parameterization with Grell 3D convective scheme (G3/CTH). Our work implements CAPE-PR lightning parameterization coupled with Kain Fritsch convective scheme into WRF-Chem, and suggests it being a better proxy for lightning in WRF-Chem. To our knowledge the CAPE-PR parameterization has not previously been coupled with chemistry.

On top of the improvement in representing lightning and lightning NO$_x$, we further investigate its effect on satellite NO$_x$ retrievals, and estimate the lightning NO$_x$ production rate by comparing modeled NO$_2$ VCD against satellite observations. Our study strongly suggests that accurately the retrieving NO$_x$ VCD requires a priori profiles produced from model simulation with reliable lightning NO$_x$. While this is not surprising our quantitive results are a useful point of reference.

Some of the discussion is rather odd. A few of the issues were raised in my initial review but no modifications were made to the paper. These issues are not complex but do require

more in-depth thinking than what was given in the manuscript. I think that the uncertainties in lightning measurements and modeling, satellite measurements and retrievals, and in situ measurements should be clearly acknowledged.

We made no scientific changes after the access review as is consistent with ACP policy. This was not meant to ignore the referee comments but rather to respond here in the open discussion.

(1) Lightning modeling uncertainty. I pointed out in the initial review that the abstract statement in line 11-12 on page 1 should be deleted. Comparing NO2 VCDs between two parameterizations is not meaningful because the amount of lightning NOx (LNOx) is a function of specified IC/CG ratio, NOx yield per flash, and the vertical distribution of LNOx. Values can be easily modified to produce similar LNOx values between the two parameterizations. For the same reason, line 12-13 on page 11 in the conclusion section should be deleted. The NOx production rate per (IC or CG) flash cannot be determined with available observations.

We agree that changing some lightning characteristics will affect $NO_2$ VCD. Our point here is emphasizing the non-negligible effect of lightning parameterizations on $NO_2$ VCD retrievals. Except for giving the exact value of changes in VCD, we modify the text in abstract to:

> "Using a lightning $NO_x$ production rate of 500 mol NO flash$^{-1}$, the a priori $NO_2$ profile generated by the simulation with the KF/CAPE-PR parameterization reduces the air mass factor for $NO_2$ retrievals by 16% on average in the southeastern US on the late spring and early summer compared to simulations using the G3/CTH parameterization. **This causes an average change in $NO_2$ vertical column density four times higher than the average uncertainty.**"

In Section 3.3, we added the following text to compare our results to the uncertainty in BEHR VCD retrievals:

> "We follow the same algorithm used in Laughner and Cohen (2017) to determine if the result is significant. The overall uncertainty due to AMF calculation for BEHR v3.0B is smaller than 30% during the study period (Laughner et al., 2019). As each grid in Fig. 3(a) is the average of 45±9 pixels, the reduced uncertainty is less than 4.5%. The overall change in VCD is four times larger than the reduced uncertainty. The switch of lightning parameterization leads to changes in VCD exceeding the averaged uncertainty in ˜94% of pixels in the southeast region of US."

We respectfully disagree with the reviewer to delete the line 12-13 on page 11. While the elements listed above affect the lightning $NO_x$ estimate, numerous studies have used the far-field approach to constrain the lightning $NO_x$ production rate with uncertainty considerably accounted for (Hudman et al., 2007; Martin et al., 2007; Jourdain et al., 2010; Miyazaki et al., 2014; Liaskos et al., 2015; Laughner and Cohen, 2017; Nault et al., 2017). To be more specific, as we assume CG and IC produce the same amount of lightning $NO_x$ per flash, and we validate the KF/CAPE-PR parameterization by comparing the total flash rate against

ENTLN, IC/CG ratio does not affect our estimate of lightning $NO_x$ production rate. For the vertical distribution of $LNO_x$, we use the modified version of profiles from Ott et al. (2010) based on the results from cloud-resolving model, which is consistent with Laughner and Cohen (2017). Overall we have optimized the affecting factors to reduce the uncertainty of lightning $NO_x$ production rate estimate using the far-field approach.

The abstract statement in line 5-6 on page 1 is based on model comparison with SEAC4RS data, when the lightning activity in the Southeast is relatively low. Unless it is a science objective, aircraft in missions like SEAC4RS usually flies in sunny days and steers away from thunderstorms. The 50% reduction is also only specific to the CAPE-PR parameterization in this paper and it offers little value to other lightning parameterizations.

The abstract statement is based on model comparison with DC3 data, referring to Section 3.1. Over the time period, the lightning activity in relatively high as DC3 is designed to observe deep convection. To avoid the confusion, we add the text in Section 3.1:

> "The lightning parameterizations are compared against observations from ENTLN in Fig. 1. Each of the datasets is averaged from May 13 to June 23, 2012, **covering DC3 field campaign.**"

The 50% reduction in the southeast region of US is applied to improve the model performance in representing lightning when it uses CAPE-PR as the lightning parameterization. We are not intended to apply this value to other lightning parameterizations. We made the following changes in the abstract for clarification:

> "We implement a lightning parameterization using the product of convective available potential energy (CAPE) and convective precipitation rate (PR) into Weather Research and Forecasting-Chemistry (WRF-Chem) model. **The CAPE-PR parameterization with a regional scaling factor of 0.5 in the southeastern US, is coupled with Kain Fritsch convective scheme (KF/CAPE-PR) to generate lightning for the continental US.** We show that.... "

It is not surprising that the CTH based parameterization doesn't work well. It's a poor choice in WRF-Chem, but I presume that it's a good reason to compare it to a new scheme. It would be better that other parameters, say those used by Allen et al. (2002) and more recently Luo et al. (2017), were used.

We agree with the reviewer that CTH based parameterization fails to reproduce good representation of lightning flashes in models. Even it is a poor choice, the CTH parameterization coupled with Grell 3D convective scheme is the only option provided by WRF-Chem to include lightning and lightning $NO_x$ into model simulation at cloud parameterized scale. Better lightning parameterizations have been discussed and evaluated in other models, for instance, WRF-CMAQ (Luo et al., 2017) and GEOS-START (Allen and Pickering, 2002). To our knowledge there is no other lightning parameterization has been implemented into WRF-Chem at convective parameterized scale. In our study, we implement CAPE-PR lightning parameterization and find it is an extraordinarily good proxy for lightning flash. While it is generally hard to compare the results across the literatures as different models, time windows

and regions are chosen, we can roughly compare our results with Luo et al. (2017), which successfully implemented most common lightning parameterizations into WRF-CMAQ. Our correlation results are better than the optimal case analyzed in Luo et al. (2017) ($R^2 = 0.56$, Slope $= 0.87$). Therefore in our study, we only implements CAPE-PR into WRF-Chem. However, in the future, we will intend to implement more lighting parameterizations studied in Luo et al. (2017) into WRF-Chem and test their performances in representing lightning and lightning $NO_x$.

A third issue raised in the initial review is line 17-19 on page 4. Which convection scheme was used for the results presented later in the paper? Are there differences? I did not find the results comparing the two convective schemes. Zhao et al. (2009) compared MM5 Grell and WRF KF schemes and found large differences. My understanding is that the Grell scheme in WRF does not have the large bias in MM5. It should be discussed.

We recognize the inconsistency in convective scheme will affect the robustness of our conclusion. We agree that the switch of convective scheme will affect the parameterized lightning flashes and as a result, we can't conclude the improvement in lightning flash representation in WRF-Chem is solely due to the new implemented CAPE-PR lightning parameterization. Therefore, we made another WRF-Chem run with Kain Frisch convective scheme and CTH lighting parameterization, referred as "KF/CTH". We add the description of this model run in Section 2.1.

"We analyze WRF-Chem outputs from three model runs. The first run, **referred as "G3/CTH"**, is consistent with Laughner and Cohen (2017); it selects the Grell 3D ensemble cumulus convective scheme (Grell, 1993; Grell and Dvnyi, 2002) and the CTH lightning parameterization. The Grell 3D convective scheme readily computes the neutral buoyancy level which serves as the optimal proxy for cloud top height (Wong et al., 2013). **The "G3/CTH" is the only option for the coupled convective-lighting parameterization used in WRF-Chem at a non-cloud resolving resolution (12 km). In addition, we run WRF-Chem with the CTH lightning parameterization coupled with the Kain-Fritsch cumulus convective scheme (Kain and Fritsch, 1990; Kain, 2004) ("KF/CTH") to test the effect of switching convective schemes. In the "KF/CTH" parameterization, the cloud top height is the level where the updraft vertical velocity equals to zero. Another run, referred as "KF/CAPE-PR",** selects the Kain-Fritsch cumulus convective scheme and the CAPE-PR lightning parameterization described above..."

In the context, we change CTH to either "G3/CTH" or "KF/CTH" and change CAPE-PR to "KF/CAPE-PR" accordingly. In Section 3.1, we added comparison of "KF/CTH" parameterized lightning flashes against ENTLN both in the text, Fig 1 and Table 1 (also labeled as Fig. R1 and Table R1 in this response).

"The G3/CTH parameterization fails to reproduce the spatial pattern of flashes observed by ENTLN over the CONUS. **Compared to the G3/CTH, the**

|              |       | G3/CTH | KF/CTH | KF/CAPE-PR |
|--------------|-------|--------|--------|------------|
| Southeastern | Slope | 2.08   | 0.94   | 0.96       |
|              | $R^2$ | 0.30   | 0.67   | 0.72       |
| Elsewhere    | Slope | 0.98   | 0.54   | 1.19       |
|              | $R^2$ | 0.27   | 0.48   | 0.62       |

Table R1: Correlation statistics between observed and modeled (G3/CTH, KF/CTH, KF/CAPE-PR) flash density per day averaged by regions

**KF/CTH parameterization improves the spatial correlation in the southeast region of US.** The KF/CAPE-PR parameterization better captures the spatial distribution of flash densities both in the southeast region and elsewhere in CONUS."

"The model using the KF/CAPE-PR lightning parameterization yields a tight correlation and slope close to the unity over the US domain. In the southeastern US, the $R^2$ increases from 0.3 to 0.7 and slope is reduced from 2.08 to 0.96 with the KF/CAPE-PR parameterization compared to the G3/CTH. **The slope for KF/CTH is comparable to KF/CAPE-PR while the $R^2$ for KF/CAPE-PR is slightly higher.**"

"Elsewhere in CONUS, **the $R^2$ for KF/CAPE-PR improves significantly to 0.6 compared to both G3/CTH and KF/CTH**. The slope for KF/CAPE-PR is 1.19, which is within the uncertainty of the detection efficiency of ENTLN. In general the KF/CAPE-PR lightning parameterization captures the day-to-day variation in flash densities better than the G3/CTH **and KF/CTH parameterizations as shown by the improved $R^2$ values.**"

The abstract is modified to:d

"**The CAPE-PR parameterization with a regional scaling factor of 0.5 in the southeastern US, is coupled with Kain Fritsch convective scheme (KF/CAPE-PR) to generate lightning for the continental US.** We show that the **KF/CAPE-PR** scheme yields an improved representation of lightning flashes in WRF when comparing against flash density from the Earth Networks Total Lightning Network. Compared to the cloud top height (CTH) lightning parameterization **coupled with Grell 3D convective scheme (G3/CTH)** used in WRF-Chem, simulated $NO_2$ profiles using the **KF/CAPE-PR** parameterization exhibit better agreement with aircraft observations in the middle and upper troposphere..."

The conclusion is also modified accordingly:

"We implement an alternative lightning parameterization based on convective available potential energy and precipitation rate into WRF-Chem **and couple**

[Figure]

Figure R1: Observed flash densities from the ENTLN dataset **(a)** and WRF-Chem using three coupled convective-lightning parameterizations, the G3/CTH parameterization **(b)**, the KF/CTH parameterization **(c)** and the KF/CAPE-PR parameterization **(d)**, respectively. The correlation of total flash density per day between WRF-Chem outputs and ENTLN for the southeastern US (denoted by the red box in **a-d**) is shown in panel **(e)** and the correlation for elsewhere in CONUS is shown in **(f)**. The model using G3/CTH is in red, KF/CTH is in green, and KF/CAPE-PR is in blue. Dash lines are corresponding fits. For slope and $R^2$, see Table 1.

**it with Kain Frisch convective scheme**. We evaluate its performance in simulating lightning $NO_x$. We first validate it by comparing against lightning observations and conclude that the **KF/CAPE-PR** parameterization with a regional scaling factor of 0.5 in the southeastern US improves...."

(2) Lighting measurement uncertainty. Section 2.2 described the ENTLN lightning data and its use in this paper. Figure 1 in the paper by Luo et al. (2017) compared the lightning distributions observed by ENTLN to NLDN. For the data they used, ENTLN had more IC and CG lightning flash rates than NLDN. The lower IC flash rates in NLDN are likely due to a lower detection efficiency. However, the CG flash rates in NLDN are also lower. As implied by Luo et al. (2017), the NLDN CG flash rate data have been the ?gold standard? in previous LNOx studies over the US. The distributions of NLDN flash rates were also different from ENTLN data in that work. The uncertainties of ENTLN data should be acknowledged. It is another reason the two statements on lightning parametrization in the abstract (discussed above) are not robust science results and should be taken out. Some explanation is due for the reasons of not using NLDN data.

While both NLDN and ENTLN have high detection efficiency (>90%) for CG flashes, we also recognize that ENTLN observes more CG flashes than NLDN. Shown in Fig. R2, we average the flashes density over CONUS both from ENTLN and NLDN between May 13 to June 23 2012. The daily averaged CG flash density from ENTLN is tightly correlated with those from NLDN with slope of 1.5. It can be explained by discrepancy in the grouping criterions applied to produce flash counts between NLDN and ENTLN. ENTLN groups all pulses within 10 km and 700 ms of each other as a single flash, and NLDN uses 10 km and 1000 ms as the threshold. In consequence, for the same amount of CG pulses measured by both lightning observation network, ENTLN produces more flashes than NLDN according to the grouping algorithm.

In our study, we use the total lightning flashes, including cloud-to-ground (CG) and Intra-cloud (IC) flashes, to validate the lightning parameterization in WRF-Chem. Compared to NLDN, ENTLN is selected for high detection efficiencies in both CG and IC flashes.The average detection efficiency for total flashes observed by ENTLN was 88% over CONUS relative to space-based TRMM-LIS (Lapierre et al., submitted) . We also evaluate ENTLN by comparing against LIS data during our study period, and the result indicates ENTLN represent total flash rates very well. We add the following text into Section 2.2:

> "Compared to National Lightning Detection Network (NLDN), ENTLN is selected for high detection efficiencies of both CG and IC flashes. The average detection efficiency for total flashes observed by ENTLN was 88% over CONUS relative to the space-based Tropical Rainfall Measurement Mission (TRMM) Lightning Imaging Sensor (LIS) (Lapierre et al. (submitted), private communication). Shown in Fig. S1, we matched the ENTLN data to LIS flashes both in time and space after the correction of LIS data based on its detection efficiency (Cecil et al., 2014) during May 13-June 23, 2012. It shows a median correlation ($R^2 = 0.51$) with the slope of 1.0, indicating the ENTLN data during the study time period is in agreement with the LIS observation. We use the ENTLN for

[Figure]

Figure R2: Comparison between CG flash density per day observed by NLDN and ENTLN. The data spans May 13 to June 23, 2012.

analysis as reported and consider the detection efficiency of ENTLN as a source of uncertainty when comparing the modeled lightning flashes."

The comparison in Fig. S1 is inadequate since it is only for one day only. LIS observations are mostly for IC flash rates and there are good reasons to believe that a CG flash produces more NOx than an IC flash (see the relevant discussion in Luo et al. (2017)).

The comparison in Fig. S1 covers May 13 to June 23, 2012, which is consistent with the study period for comparing WRF-Chem lightning parameterizations against ENTLN. TRMM-LIS has been shown to observe CG and IC flashes equally well and the detection efficiency varies between 0.69 to 0.88 by hour of the day. To better explain Fig. S1, we add the following text in Section 2.2.

"Shown in Fig. S1, we matched the ENTLN data to LIS flashes both in time and space after the correction of LIS data based on its detection efficiency (Cecil et al., 2014) during May 13-June 23, 2012. It shows a median correlation ($R^2 = 0.51$) with the slope of 1.0, indicating the uncorrected ENTLN data during the study time period shows the best agreement with LIS observation."

We also add more detailed description in the caption of Fig. S1 (also labeled as Fig. R3 in this response):

[Figure]

Figure R3: Comparison between flash rates observed by ENTLN and Lightning Imaging Sensor (LIS). **ENTLN data is matched to corrected LIS flashes both in time and space during May 13-June 23, 2012, and both datasets are summed onto 0.5°x 0.5°grid spacing.** (a,b) shows the spatial pattern of lightning flash rates measured by LIS (a) and ENTLN (b). The plot region covers 20°N - 38°N and 130°W - 65°W. (c,d) are corresponding absolute difference and scatter plots between LIS and ENTLN. **LIS data is corrected using the detection efficiency from** Cecil et al. (2014).

The difference between CG and IC flashes relative to $NO_x$ production rate is disputable. We assume CG and IC flashes produce the same amount of $NO_x$ per flash in this study primarily due to two reasons:

1. Among literatures suggesting CG flash produces more $NO_x$ than an IC flash, there is no agreement on quantifying the difference (Koshak et al., 2014; Carey et al., 2016; Luo et al., 2017; Lapierre et al., submitted).

2. The disputation is partially due to the ambiguity in the concept of lightning flash. By definition, a flash contains multiple strokes, and strokes are impulsive current pulses measured directly by the sensors configured in the lightning observation network. A lightning flash is classified as CG when it contains a return stroke, otherwise it is classified as IC. Lapierre et al. (submitted) indicates that a CG stroke produces much more (10 times) $LNO_x$ than a IC stroke, of which the conclusion is consistent with (Koshak et al., 2014). However, $LNO_x$ derived from flash rather than stroke will obscure the IG and CG variability as a IC flash contains more strokes than a CG flash.

(3) OMI retrieval uncertainties. The comparisons of Figures 4 and S2 can only be used as a qualitative not quantitative measure of lightning NOx in the model. For clean regions of the SE, the difference is on the order of 1x1015 molec cm-2. Considering that the OMI uncertainty is larger than 1.5x1015 molec cm-2, it is difficult to say which sensitivity simulation is quantitatively better. There are additional uncertainties in OMI retrievals including surface albedo, cloud, and background noise. With the uncertainties in mind, the difference among the lightning sensitivity simulations in Figure S3 is therefore insignificant. Appropriate discussion on the uncertainties of OMI retrievals and the implications for this paper should be included.

We appreciate reviewer's suggestion on discussing the uncertainty of OMI retrievals. However, we believe the results we shown in the manuscript is significant. The results represent the average of $NO_2$ VCD between May 13 to June 23, 2012, and $32 \pm 6$ pixels contribute to each value. While the global mean uncertainty for tropospheric $NO_2$ VCD retrievals is $1 \times 10^{15}$ mole cm$^{-2}$ (Bucsela et al., 2013), the reduced uncertainty in our analysis is $\sim 0.2 \times 10^{15}$ mole cm$^{-2}$, which is less than half of the RMSEs we calculated between BEHR retrievals and model simulations. We expand the description of the figures in the discussion:

> "Figure 4 shows the difference between satellite retrieved $NO_2$ VCD and model simulated $NO_2$ VCD without lightning $NO_x$ **(a)** and with lightning $NO_x$ production rate of 500 mol NO flash$^{-1}$ **(b)** averaged between May 13 to June 23, 2012. Figure S2 shows difference plots with varied lightning $NO_x$ production rates (400 and 665 mol NO flash$^{-1}$). The corresponding root-mean-square errors (RMSE) are included in Table S1. $LNO_x$ production rate of 500 mol NO flash$^{-1}$ yields the lowest RMSE of $0.41 \times 10^{15}$ mole cm$^{-2}$ between modeled and observed $NO_2$ VCD over CONUS. This is at the high end of previous estimates of the lightning $NO_x$ production rate (16-700 mol NO flash$^{-1}$). "

> "The RMSE for urban areas (top 5% of $NO_2$ VCD simulated by WRF-Chem without $LNO_x$) remains at high value ($\sim$0.9-1.3$\times 10^{15}$ mole cm$^{-2}$) when switching

the LNO$_x$ production rate. It indicates that the bias in the modeled VCD over urban areas is more likely due to surface NO$_2$. The RMSE for non-urban areas shows pronounced change with varied LNO$_x$ production rate. Excluding urban areas lowers the RMSE to $0.37 \times 10^{15}$ mole cm$^{-2}$ for LNO$_x$ production rate of 500 mol NO flash$^{-1}$. The RMSEs are significant considering the uncertainty for retrievals. During the average time period, $32 \pm 6$ pixels contribute to each value in the plots. While the global mean uncertainty for tropospheric NO$_2$ VCD retrievals is $1 \times 10^{15}$ mole cm$^{-2}$ (Bucsela et al., 2013), the reduced uncertainty in our analysis is $\sim 0.2 \times 10^{15}$ mole cm$^{-2}$. The calculated RMSEs are twice of the uncertainty. ”

There is no description on how OMI data under high cloud-fraction conditions are treated. Those data cannot be used in the comparisons of Figures 4 and S2.

The pixels with cloud fraction larger than 0.2 are filtered out in our analysis. We add the text in the caption of Fig.4 for clarification.

"**Figure 4.** Difference in NO$_2$ VCD between BEHR retrievals and WRF-Chem. **(a)** excludes LNO$_x$ in model simulation, **(b)** adds LNO$_x$ emission with production rate of 500 mol NO flash$^{-1}$. **(c)** includes the same LNO$_x$ emission as **(b)** but uses NO$_2$ profiles scaled upward by 60% at pressure lower than 400 hPa. **The average time covers May 13 to June 23, 2012. Pixels with cloud fraction larger than 0.2 are filtered out in the analysis.**"

(4) Uncertainty of the upper tropospheric NO2/NOx ratio. In sections 4, the uncertainty of the upper tropospheric NO2/NOx ratio was discussed. This issue doesn?t affect model comparison with in situ observations when NO measurements are available. Comparisons with in situ NO, O3, and JNO2 should be included with the discussion of Figure 2. To evaluate model lightning NOx simulations, using NO measurements will get around the uncertainty of NO2/NOx ratio. Therefore, the last statement of the conclusion section (line 13-14 on page 11) is inappropriate and should be removed. This uncertainty affects the retrieval of OMI data only if the unknown interferences by other nitrates are insignificant. Furthermore, the uncertainties oof OMI data may mask out the effects. More detailed discussion should be included.

While NO measurements are available during DC3 and SEAC4RS field campaigns, we disagree that the uncertainty of NO$_2$/NO$_x$ ratio will not affect the model results. Our results are consistent with Silvern et al. (2018). In Figure 1 from Silvern et al. (2018), they compared the profiles of NO/NO$_2$ and relative quantities on SEAC4RS flights and concluded there is no systematic model bias in ozone, temperature, or JNO$_2$ that would explain the error in NO/NO$_2$.

The incorrect nitrate chemistry forming PNs/ANs/HNO$_3$ will affect the result, however, we argue the error from the underestimated NO$_2$/NO$_x$ ratio is still significant:

1. We use a customized version of the Regional Atmospheric Chemistry Mechanism version 2 (RACM2), the details are described by Zare et al. (2018), which has a very

detailed nitrate mechanism, so any errors should be smaller than in most model comparisons.

2. Nault et al. (2017) found a 33% error in upper tropospheric $NO_2$ caused by the incorrect nitrate chemistry before the modified mechanism is implemented. In this study, if we assume that most of the $LNO_x$ falls in pressure lower than 400 hPa, then given that +60% $NO_2$ is too much $NO_2$ in the column and the +0% was too little, a ballpark estimate of ~30% seems reasonable. That's the same order as Nault et al. (2017) saw for nitrate chemistry, so the $NO_2/NO_x$ ratio can be significant even if nitrate chemistry is poorly constrained.

(1) P. 4, Line 26-27, Eq. (2), Luo et al. (2017) used a formula of $f = a_0 \times x^{a_1} + a_2 \times x \times y + a_3 \times y^{a_4}$ , what is the reason for not including the other terms? What are the reasons for not using UMF or CPR?

The formula used in Luo et al. (2017) is based on the power law relationship. The CAPE-PR lightning parameterization used in the paper is modified based on Romps et al. (2014), from which they found tight relationship between observed lightning flash rate and CAPE times precipitation rates both from measurements.

As we mentioned above, except for CTH, neither UMF or CPR has been implemented into WRF-Chem. Our study implements CAPE-PR lightning parameterization into WRF-Chem and find out that it reproduce lightning flashes well comparing against lightning observation. In the future, more lightning parameterizations will be implemented to WRF-Chem in order to further improve its performance in representing lightning and lightning $NO_x$.

(2) P. 8, Figure 3, Is the large urban VCD change mostly due to Orlando? Figure 1 shows very little lightning activity in the other SE region.

Figure 1 and Figure 3 correspond to different time periods. Figure 1 shows the spatial distribution of lightning flash density averaged from May 13 to Jun 23, 2012 covering DC3; Figure 3 shows the change of VCD averaged from Aug 01 to Sep 23, 2013 as Figure 2 shows large difference in $NO_2$ profiles during SEAC4RS. As ENTLN is only available upon request, we currently have no ENTLN data covering the same time period as Figure 3 . However, we find out the Figure 3 (b) remain unaffected if we mask out the Florida area, thus the large urban VCD is not due to Orlando.

(3) P. 9, Line 6-7, the 19% value should be compared to Zhao et al. (2009).

Zhao et al. (2009) and this study are looking at different quantities. Zhao et al. (2009) discussed about how much lightning contributes to a modeled $NO_2$ column, while this study is looking at how the change in a priori profiles from different lightning parameterizations affects a retrieved VCD.

(4) Figures are hard to read in general.

We enlarge all figures within the space constraints of the ACP template.

[revised manuscript text omitted]
). ENTLN data is matched to corrected LIS flashes both in time and space during May 13-June 23, 2012, and both datasets are summed onto 0.5°x 0.5°grid spacing. (a,b) shows the spatial pattern of lightning flash rates measured by LIS (a) and ENTLN (b). The plot region covers 20°N - 38°N and 130°W - 65°W. (c,d) are corresponding absolute difference and scatter plots between LIS and ENTLN. LIS data is corrected using the detection efficiency from citetcecil14.

Comparison between flash rates observed by ENTLN and Lightning Imaging Sensor (LIS). (a,b) shows the spatial pattern of lightning flash rates averaged from May 13 to Jun 23 2012 measured by LIS (a) and ENTLN (b). The plot region covers 20°N - 38°N and 110°W - 65°W. (c,d) are corresponding absolute difference and scatter plots between LIS and ENTLN.

| | No lightning | 400 mol NO $flash^{-1}$ | 500 mol NO $flash^{-1}$ | 665 mol NO $flash^{-1}$ |
|---|---|---|---|---|
| CONUS | $0.92 \times 10^{15}$ | $0.44 \times 10^{15}$ | $0.41 \times 10^{15}$ | $0.44 \times 10^{15}$ |
| Urban | $1.30 \times 10^{15}$ | $0.89 \times 10^{15}$ | $0.91 \times 10^{15}$ | $1.10 \times 10^{15}$ |
| Non-Urban | $0.90 \times 10^{15}$ | $0.41 \times 10^{15}$ | $0.37 \times 10^{15}$ | $0.39 \times 10^{15}$ |

Table S1: The root-mean-square errors (RMSE) in unit of mole $cm^{-2}$ between observed and modeled $NO_2$ VCD using WRF-Chem with varied $LNO_x$ production rates (0, 400, 500, 665 mol NO $flash^{-1}$). Urban areas are selected where $NO_2$ columns are at top 5% calculated from WRF-Chem without lightning. Non-urban areas are CONUS excluding urban areas.

Box plot of difference in $NO_2$ VCD between BEHR retrievals and WRF-Chem with varied $LNO_x$ production rate of 0, 400, 500 and 665 mol NO flash$^{-1}$. The corresponding root-mean-square errors (RMSE) are shown above.

[Figure]

Figure S2: Difference in $NO_2$ VCD between BEHR retrievals and WRF-Chem **(a)** without $LNO_x$ and with $LNO_x$ production rate of **(b)** 400 mol NO flash$^{-1}$, **(c)** 500 mol NO flash$^{-1}$ and **(d)** 665 mol NO flash$^{-1}$.

[Figure]

Figure S3: Comparison of WRF-Chem and aircraft $[NO_2/NO_x]$ profiles from the **(a)** DC3, **(b)** SEAC4RS campaigns.The solid line is the median of all profiles and the shaded areas are between 10th and 90th percentiles for each binned level. Aircraft measurements are shown in black, WRF-Chem using CTH lightning parameterization in red and WRF-Chem using CAPE-PR lightning parameterization in blue.

---

## Author Response (AR2)

**Lightning NO$_2$ simulation over the Contiguous US and its effects on satellite NO$_2$ retrievals**

**Response to Editor**

Qindan Zhu, Joshua L. Laughner and Ronald C. Cohen

August 22, 2019

We thank the editor for very careful reading of both the main article and the supplement. Below we respond to the individual comments. The reviewer's comments will be shown in red, our response in blue, and changes made to the paper are shown in black block quotes. Unless otherwise indicated, page and line numbers correspond to the original paper. Figures, tables, or equations referenced as "R$n$" are numbered within this response; Figures, tables, and equations numbered normally refer to the numbers in the original discussion paper.

In Figure 3, is the cloud fraction threshold of 0.2 used as in Figure 4? If so, please add this information to the caption

The cloud fraction threshold of 0.2 is also used in Figure 4. We modified the caption of Figure 4:

> "Relative change in BEHR NO$_2$ VCD over the southeastern US switching the source of a prior NO$_2$ profiles from WRF-chem outputs using G3/CTH to one using KF/CAPE-PR lightning parameterization. **(a)** shows the mean spatial distribution of the changes from Aug 01 to Sep 23, 2013 and **(b)** shows the temporal variation over urban and rural areas. **Only observations with cloud fraction less than 20% are included.** Medium to large cities, including Atlanta, GA; Huntsville, AL; Birmingham, AL; Tallahassee, FL; Orlando, FL; and Baton Rouge, LA, are marked by stars in panel **(a)**."

On page 10, line 2, you cite Laughner et al., 2019 for the statement that "uncertainty due to AMF calculation for BEHR v3.0B is smaller than 30%. I couldn't find the corresponding statement in that paper and think that more qualification is needed? which parts of the uncertainty do you consider in this number? The way this number is used suggests that you exclude uncertainty from the vertical profile used? Why is it OK to assume that AMF uncertainties (which are strongly related to knowledge of surface reflection) are reduced like a random uncertainty when averaging?

The uncertainty evaluation of BEHR v3.0B is in the Section 6 of supplementary from Laughner et al. (2019). Summarized in Table S4 from Laughner et al. (2019), the uncertainty in AMF due to surface reflectance, surface pressure, tropopause pressure, cloud pressure,

cloud radiance fraction, and a priori profiles is determined by perturbing each parameter and re-retrieving the $NO_2$ VCD with the perturbed values. The calculated AMF uncertainty is less than 30% except for the winter.

Note that the uncertainty from the vertical profile is also included in the estimated AMF uncertainty. By improving the lightning parameterization in the models, we expect the uncertainty from the vertical profiles is lower than the previous calculation. The uncertainty in AMF of 30% is a very conservative estimate.

According to Figure S12 in Laughner et al. (2019), the a priori profile is the largest contributor to the AMF uncertainty, and tropopause pressure and cloud pressure are the next two largest contributors. Given that the uncertainty due to surface reflection is very small in general ($<4\%$), we can treat daily AMF and VCD as independent variables and calculate the uncertainty duo to AMF calculation as random uncertainty. We add the following text to point out the sources of uncertainty:

> "We follow the same algorithm used in Laughner and Cohen (2017) to determine if the result is significant. The overall uncertainty due to AMF calculation for BEHR v3.0B is smaller than 30% during the study period **(Sec 6 in supplementary from Laughner et al. (2019)).Over 90% of the uncertainty attributes to the a prior $NO_2$ profiles, the tropopause and cloud pressures.** As each grid in Fig. 3(a) is the average of 45±9 pixels, the reduced uncertainty is less than 4.5%."

In order to better understand the reason for the large changes over urban areas when changing the lightning parametrisation it would be good to add figures showing vertical profiles of NO2 for the three cases of no lightning, old parametrisation and new parametrisation as well as the scattering weights for the two cases of August 24 and September 10 that you discuss

We add Fig. S3 (also labeled as Fig. R1 in this response) in the supplementary. Note that we did not include the $NO_2$ profiles from WRF-Chem without lightning $NO_x$ emissions, which are similar to the $NO_2$ profiles from WRF-Chem using KF/CAPE-PR with little $NO_2$ in the middle and upper troposphere over both urban and rural areas. For both days, switching from G3/CTH to KF/CAPE-PR parameterization in WRF-Chem substantially lowers the $NO_2$ in the upper troposphere. The difference in the relative change of VCD between two days is mainly due to the sensitivity in AMF to the erroneous high peak of $NO_2$ caused by G3/CTH parameterization in the middle and upper troposphere. We expand the discussion of the VCD changes on Sep 10 and Aug 24:

> "Table 2 presents the AMF and VCD obtained from using a priori profiles with G3/CTH or KF/CAPE-PR lightning parameterizations as well as the relative changes on Sep 10 and Aug 24, 2013. **The corresponding a priori $NO_2$ profiles and scattering weights over urban and rural areas are shown in Fig. S3.** Sep 10 is an example of one day when the change in $NO_2$ profiles has a very large impact on the $NO_2$ VCDs. **The WRF-Chem using G3/CTH parameterization places a large amount of $NO_2$ between 200-600 hPa with the maximum value comparable to the near surface $NO_2$ over the**

**urban areas. The calculated AMF is predominantly determined by lightning $NO_2$ due to the combination of higher scattering weight and larger $NO_2$ in the middle and upper troposphere.** The change in AMF is -56.0% over urban areas and -32.0% over rural areas; the corresponding VCD increases by 134.9% and 44.9%, respectively. In contrast, Aug 24 is an example where the lightning parameterization has very little effect. **While the positive bias in $NO_2$ aloft is also observed by using G3/CTH parameterization, the amount of $NO_2$ in the middle and upper troposphere is smaller than Sep 10. It leads to lower sensitivity in AMF to the erroneous $NO_2$ caused by the lightning parameterization. With smaller relative change in AMF,** the relative change in VCD is 3.1% over urban areas and -4.6% over rural areas. "

Please change the labels in Fig. 4 from "change in" to "difference in"
The figure is modified accordingly.

Please check again if the figures really show BEHR-WRF-Chem as stated in the text and in the caption. Judging from the numbers, I would guess that in fact WRF-Chem - BEHR is shown.
The figure shows WRF-Chem minus BEHR. We add it into the caption:

"Difference in $NO_2$ VCD between BEHR retrievals and WRF-Chem (**"WRF-Chem" − "BEHR"**). **(a)** excludes $LNO_x$ in model simulation, **(b)** adds $LNO_x$ emission with production rate of 500 mol NO flash$^{-1}$. **(c)** includes the same $LNO_x$ emission as **(b)** but uses $NO_2$ profiles scaled upward by 60% at pressure lower than 400 hPa. The average time covers May 13 to June 23, 2012. Pixels with cloud fraction larger than 0.2 are filtered out in the analysis."

Why is the spatial pattern of lightning (Fig. 1d) not reflected in Figure 4a?
Figure 1 and 4 average the datasets during the same study period. However, Fig 1 shows the daily lightning density over the US domain, whereas Fig 4 shows the $NO_2$ VCD at the OMI overpass time ($\sim$1:30 pm local time). Only lightning occurring before OMI overpass time contributes to the observed $NO_2$ VCD. We add it in Sec 2.4:

"The Ozone Monitoring Instrument (OMI) is an ultraviolet/visible (UV/Vis) nadir solar backscatter spectrometer launched in July 2004 on board the Aura satellite. It detects backscattered radiance in the range of 270-500 nm and the spectra are used to derive column $NO_2$ at a spatial resolution of 13 km$\times$24 km at nadir (Levelt et al., 2006). **The OMI overpass time is $\sim$13:30 local time.**"

**References**

Laughner, J. L. and Cohen, R. C.: Quantification of the effect of modeled lightning $NO_2$ on UV–visible air mass factors, Atmospheric Measurement Techniques, 10, 4403–4419, doi:

[Figure]

Figure R1: The a priori NO$_2$ vertical profiles **(a, b)** and scattering weights **(c, d)** on Sep 10 and Aug 24 2013 over all urban (solid) or rural (dashed) grid cells in SE US. The NO$_2$ profiles from WRF-Chem using G3/CTH parameterization are in red, those from KF/CAPE-PR parameterization are in blue.

10.5194/amt-10-4403-2017, URL https://www.atmos-meas-tech.net/10/4403/2017/, 2017.

Laughner, J. L., Zhu, Q., and Cohen, R. C.: Evaluation of version 3.0B of the BEHR OMI $NO_2$ product, Atmospheric Measurement Techniques, 12, 129–146, doi:10.5194/amt-12-129-2019, URL https://www.atmos-meas-tech.net/12/129/2019/, 2019.

Levelt, P., Oord, G., R. Dobber, M., Mlkki, A., Visser, H., Vries, J., Stammes, P., Lundell, J., and Saari, H.: The Ozone Monitoring Instrument, IEEE T. Geoscience and Remote Sensing, 44, 1093–1101, doi:10.1109/TGRS.2006.872333, 2006.

**Lightning NO$_2$ simulation over the Contiguous US and its effects on satellite NO$_2$ retrievals**

**Response to Anonymous Referee #1**

Qindan Zhu, Joshua L. Laughner and Ronald C. Cohen

August 22, 2019

We thank the reviewer for the positive response of the main article. Below we respond to the individual comments. The reviewer's comments will be shown in red, our response in blue, and changes made to the paper are shown in black block quotes. Unless otherwise indicated, page and line numbers correspond to the original paper. Figures, tables, or equations referenced as "R$n$" are numbered within this response; Figures, tables, and equations numbered normally refer to the numbers in the original discussion paper.

The abstract, the results, and the conclusions need to be enhanced to indicate the relative amounts of improvement due to changing convective schemes and changing lightning schemes. Table 1 shows that in the southeastern US just changing from G3 to KF produces the bulk of the improvement in slope and R2. With the KF scheme, changing from CTH to CAPE-PR only makes a small incremental improvement in slope and R2. Elsewhere, the change from G3/CTH to KF/CTH makes a 50% greater improvement in R2 than changing from KF/CTH to KF/CAPE-PR. So, the bottom line is is that the convective scheme was more important than lightning scheme in yielding improved lightning prediction. The paper needs to say this.

Thanks for the suggestion. We modify the abstract to:

> "Lightning is an important NO$_x$ source representing ~10% of the global source of odd N and a much larger percentage in the upper troposphere. The poor understanding of spatial and temporal patterns of lightning contributes to a large uncertainty in understanding upper tropospheric chemistry. **We implement a lightning parameterization using the product of convective available potential energy (CAPE) and convective precipitation rate (PR) coupled with Kain Fritsch convective scheme (KF/CAPE-PR) into Weather Research and Forecasting-Chemistry (WRF-Chem) model. Compared to the cloud top height (CTH) lightning parameterization combined with Grell 3D convective scheme (G3/CTH), we show that the switch of convective scheme improves the correlation of lightning flash density in the southeastern US from 0.30 to 0.67 when comparing against the Earth Networks Total Lightning Network; the switch**

**of lightning parameterization contributes to the improvement on correlation from 0.48 to 0.62 elsewhere in the US.** The simulated $NO_2$ profiles using the KF/CAPE-PR parameterization exhibit better agreement with aircraft observations in the middle and upper troposphere......"

In the Section 3.1, we rewrite the discussion on the comparison between modeled and observed lightning flash densities:

"Both models using the KF/CTH and KF/CAPE-PR parameterizations improves the correlation between modeled and observed lightning flash densities over the US domain. In the southeastern US, changing from G3 to KF convective scheme substantially increases the $R^2$ from 0.30 to 0.67 and reduces the slope from 2.08 to 0.94. Switching from CTH to CAPE-PR lightning parameterization only contributes a slight increment on the correlation. While the slopes close to unity both for KF/CTH and KF/CAPE-PR, we note that the improved scaling of the slope in KF/CAPE-PR is mainly caused by the scaling factor of 0.5 applied to the southeast region. In this simulation, a constant linear coefficient for CAPE-PR is not adequate to represent the observed lightning over CONUS, in contrast to the finding of Romps et al. (2014). Elsewhere in CONUS, both the changes in convective scheme and lightning parameterization yield a better representation of lightning flash densities compared to the observation. The $R^2$ for KF/CAPE-PR improves significantly to 0.62 compared to both G3/CTH and KF/CTH. The slope for KF/CAPE-PR is 1.19, which is within the uncertainty of the detection efficiency of ENTLN. In general the KF/CAPE-PR lightning parameterization captures the day-to-day variation in flash densities better than the G3/CTH and KF/CTH parameterizations as shown by the improved $R^2$ values."

The conclusion is also modified accordingly:

"We implement an alternative lightning parameterization based on convective available potential energy and precipitation rate into WRF-Chem and couple it with Kain Frisch convective scheme. **We first validate it by comparing against lightning observations and find that the switch of convective scheme reproduces day-to-day variation of lightning flashes in the southeastern US and the switch of lightning parameterization contributes to the improvement on lightning representation elsewhere in the US.** We also compare the simulated $NO_2$ profiles against aircraft measurements and find that the simulated $NO_2$ using KF/CAPE-PR is more consistent with observations in the mid and upper troposphere."

Figure 3 shows the VCD changes by switching a priori profiles produced from WRF-Chem using G3/CTH to the one using KF/CAPE-PR. Since this is a model-model comparison the accuracy of the lightning observations is not especially useful. The area with large VCD difference is also much larger than Orlando from Fig 3 (a).

Besides, the VCD changes are not linear with lightning changes in the models. VCD is calculated by dividing SCD (slant column density) by AMF, and AMF is obtained based on scattering weights and a priori $NO_2$ profiles. We add a more detailed discussion on the AMF and VCD changes in the last paragraph of Sec 3.3:

[revised manuscript text omitted]

**S1  Comparison between ENTLN and NLDN**

While both NLDN and ENTLN have high detection efficiency (>90%) for CG flashes, we recognize that ENTLN observes more CG flashes than NLDN. Shown in Fig. **??**, we average the flashes density over CONUS both from ENTLN and NLDN between May 13 to June 23 2012. The daily averaged CG flash density from ENTLN is tightly correlated with those from NLDN with slope of 1.5. It can be explained by discrepancy in the grouping criterions applied to produce flash counts between NLDN and ENTLN. ENTLN groups all pulses within 10 km and 700 ms of each other as a single flash, and NLDN uses 10 km and 1000 ms as the threshold. In consequence, for the same amount of CG pulses measured by both lightning observation network, ENTLN produces more flashes than NLDN according to the grouping algorithm.

[Figure]

Figure S1: Comparison between CG flash density per day observed by NLDN and ENTLN. The data spans May 13 to June 23, 2012.

[Figure]

Figure S2: Comparison between flash rates observed by ENTLN and Lightning Imaging Sensor (LIS). ENTLN data is matched to corrected LIS flashes both in time and space during May 13-June 23, 2012, and both datasets are summed onto 0.5°x 0.5°grid spacing. **(a,b)** shows the spatial pattern of lightning flash rates measured by LIS **(a)** and ENTLN **(b)**. The plot region covers 20°N - 38°N and 130°W - 65°W. **(c,d)** are corresponding absolute difference and scatter plots between LIS and ENTLN. LIS data is corrected using the detection efficiency from citetcecil14.

[Figure]

Figure S3: The a priori NO₂ vertical profiles (a, b) and scattering weights (c, d) on Sep 10 and Aug 24 2013 over urban and rural areas in SE US. The NO₂ profiles from WRF-Chem using G3/CTH parameterization are in red, those from KF/CAPE-PR parameterization are in blue.

|            | No lightning        | 400 mol NO $flash^{-1}$ | 500 mol NO $flash^{-1}$ | 665 mol NO $flash^{-1}$ |
|------------|---------------------|-------------------------|-------------------------|-------------------------|
| CONUS      | $0.92\times10^{15}$ | $0.44\times10^{15}$     | $0.41\times10^{15}$     | $0.44\times10^{15}$     |
| Urban      | $1.30\times10^{15}$ | $0.89\times10^{15}$     | $0.91\times10^{15}$     | $1.10\times10^{15}$     |
| Non-Urban  | $0.90\times10^{15}$ | $0.41\times10^{15}$     | $0.37\times10^{15}$     | $0.39\times10^{15}$     |

Table S1: The root-mean-square errors (RMSE) in unit of mole cm$^{-2}$ between observed and modeled NO$_2$ VCD using WRF-Chem with varied LNO$_x$ production rates (0, 400, 500, 665 mol NO $flash^{-1}$). Urban areas are selected where NO$_2$ columns are at top 5% calculated from WRF-Chem without lightning. Non-urban areas are CONUS excluding urban areas.

[Figure]

Figure S4: Difference in NO$_2$ VCD between BEHR retrievals and WRF-Chem **(a)** without LNO$_x$ and with LNO$_x$ production rate of **(b)** 400 mol NO flash$^{-1}$, **(c)** 500 mol NO flash$^{-1}$ and **(d)** 665 mol NO flash$^{-1}$.

[Figure]

Figure S5: Comparison of WRF-Chem and aircraft $[NO_2/NO_x]$ profiles from the **(a)** DC3, **(b)** SEAC4RS campaigns.The solid line is the median of all profiles and the shaded areas are between 10th and 90th percentiles for each binned level. Aircraft measurements are shown in black, WRF-Chem using CTH lightning parameterization in red and WRF-Chem using CAPE-PR lightning parameterization in blue.

---

## Author Response (AR3)

**Lightning NO$_2$ simulation over the Contiguous US and its effects on satellite NO$_2$ retrievals**

**Response to Editor**

Qindan Zhu, Joshua L. Laughner and Ronald C. Cohen

September 29, 2019

We thank the editor for further comments. Below we respond to the individual comments. The reviewer's comments will be shown in red, our response in blue, and changes made to the paper are shown in black block quotes.

"...and find that the switch of convective scheme reproduces day-to-day variation ..." - it's not the switch that reproduces variations but rather the model which reproduces variation after switching the convective scheme

We changed the sentence in the conclusion:

> "We first validate it by comparing against lightning observations and find that **the model reproduces day-to-day variation of lightning flashes in the southeastern US after the switch of convective scheme** and..."

I think it would also be worthwhile to state somewhere in the text or at the new figure S4 that from your work, there is (at least in my understanding) no indication that lightning NO2 has a significant impact on the AMF at the time and location you are investigating. In your conclusions, you state that "This study emphasizes the importance of including reliable lightning NO2 in a priori profiles for satellite retrievals" but the way I see it, just switching off lightning in the model would probably lead to AMF values comparable to those you determined with the updated version of the model. This is why I suggested to include the NO2 model without lightning in Fig. S4, just to demonstrate that lightning (if modelled correctly) is not relevant here.

We add it in the last paragraph of Section 3.3:

> "The corresponding a priori NO$_2$ profiles and scattering weights over urban and rural areas are shown in Fig. S3. **The G3/CTH parameterization has substantially more lightning than observed and thus places a large fraction of NO$_2$ in the upper troposphere whereas the KF/CAPE-PR has less lightning and is more consistent with observations. The resulting profiles of modeled NO$_2$ are more dominated by boundary layer NO$_2$ and less sensitive to lightning.** "

"This study indicates that the erroneous representation of lightning $NO_2$ in a priori profiles is an important source of bias for satellite retrievals."